# Improving the hydrological consistency of a process-based solute-transport model by simultaneous calibration of streamflow and stream concentrations

Jordy Salmon-Monviola[1], Ophélie Fovet[1], Markus Hrachowitz[2]

[1] UMR SAS, INRAE, Institut Agro, Rennes, France

[2] Department of Water Management, Faculty of Civil Engineering and Geosciences, Delft University of Technology, Stevinweg 1, 2628CN Delft, Netherlands

*Corresponding author*: J. Salmon-Monviola (jordy.salmon-monviola@inrae.fr)

**Abstract.** The consistency of hydrological models, i.e. their ability to reproduce observed system dynamics, needs to be improved to increase their predictive power. As using streamflow data alone to calibrate models is not sufficient to constrain them and render them consistent, other strategies must be considered, in particular using additional types of data. The aim of this study was to test whether simultaneous calibration of dissolved organic carbon (DOC) and nitrate ($NO_3^-$) concentrations along with streamflow improved the hydrological consistency of a parsimonious solute-transport model. A multi-objective approach with four calibration scenarios was used to evaluate the model's predictions for an intensive agricultural headwater catchment. After calibration, the model reasonably reproduced simultaneously the dynamics of discharge and DOC and $NO_3^-$ concentrations in the stream of the headwater catchment from 2008-2016. Evaluation using independent datasets indicated that the model usually reproduced dynamics of groundwater level and soil moisture in upslope and riparian zones correctly for all calibration scenarios. Using daily stream concentrations of DOC and $NO_3^-$ along with streamflow to calibrate the model did not improve its ability to predict streamflow for calibration or evaluation periods. The approach improved significantly the representation of groundwater storage and to a lesser extent soil moisture in the upslope zone but not in the riparian zone. Parameter uncertainty decreased when the model was calibrated using solute concentrations, except for parameters related to fast and slow reservoir flow. This study shows the added value of using multiple types of data along with streamflow, in particular DOC and $NO_3^-$ concentrations, to constrain hydrological models to improve representation of internal hydrological states and flows. With the increasing availability of solute data from catchment monitoring, this approach provides an objective way to improve the consistency of hydrological models that can be used with confidence to evaluate scenarios.

**Keywords**: Hydrological models, Equifinality, Consistency, Multi-objective calibration, Stream DOC and nitrate concentrations, parsimony.

## 1. Introduction

Hydrological models are important tools for short-term forecasting of river flows and long-term predictions for strategic water management planning, as well as for improving understanding of hydrological processes and the complex interactions of water storage and release processes at the catchment scale (Bouaziz et al., 2021; Lan et al., 2020; Minville et al., 2014). In the wide spectrum of modelling, which ranges from simple to complex (Adeyeri et al., 2020; Gharari et al., 2014; Hrachowitz and Clark, 2017), conceptual models, in which only the dominant processes are represented and/or several processes may be lumped into a single expression (Pettersson et al., 2001), are widely used to simulate hydrological dynamics of catchments. Conceptualising the system as a set of storage components connected by flows representing the perceived dominant processes of a catchment provides a certain degree of flexibility. The ability to customize these models to the environmental conditions in a given catchment can ensure an appropriate level of complexity to reproduce response patterns of hydrology and water quality (Hrachowitz et al., 2016). Major advantages of conceptual models include their relatively low data and computational requirements, which makes them suitable for studies at different scales or for catchments about which little information is available (Gharari et al., 2014; Huang and Bardossy, 2020). However, ad hoc implementation of conceptual models frequently lacks a plausible theoretical basis and thus a meaningful connection of model structure and parameters to observable quantities when representing integrated system processes (Clark et al., 2016). As such, the ability of models, including conceptual ones, to reproduce a system's dynamics is also undermined not only by random uncertainties in the data, but also by epistemic or ontological uncertainties and thus by limited knowledge of the physical processes that underlie the system's response (Beven, 2013; Beven and Westerberg, 2011; Gupta et al., 2012). These uncertainties and the few observations in a continuous spatial domain make such models ill-posed inverse problems (Beven, 2006; Hrachowitz et al., 2014; Pettersson et al., 2001). In hydrology, frequently referred to as equifinality (Beven, 2006), these insufficient model constraints thus result in many, equally good alternative model solutions. Hydrological models with many parameters thus tend to adapt to errors and to compensate for inadequate representation of processes through the model parameters (Wang et al., 2012). For example, well-predicted river discharge is often associated with poorly predicted evaporation flows, because evaporation compensates for errors and closes the hydrological balance (Minville et al., 2014). Thus, deceptively high calibration accuracy may reflect mathematical fitting of an often overparameterized model, which may generate undesirable internal dynamics that decrease accuracy in independent evaluation periods (Fovet et al., 2015a; Hrachowitz et al., 2014). Robust model calibration and evaluation procedures are thus needed to address issues of parameter identifiability (Beven, 2006; Guillaume et al., 2019) and transferability (Hartmann and Bárdossy, 2005; Kreye et al., 2019; Minville et al., 2014), and to avoid models that act as "mathematical marionettes" dancing to match the calibration data (Kirchner, 2006) but often fail to reproduce internal system dynamics.

Recently, a trend toward more comprehensive assessment of the structural adequacy of models has emerged during the calibration process (Rakovec et al., 2016; Yen et al., 2014), with the overall goal of improving the representation of multiple hydrological processes in a model (Clark et al., 2011; Euser et al., 2015; Gupta et al., 2012). The rationale behind this goal is the need to obtain the "right answers for the right reasons" (Blöschl, 2001; Kirchner, 2006), which goes beyond simply comparing model predictions to observed streamflow or associated signature measurements (Euser et al., 2013; Fovet et al., 2015a; Rakovec et al., 2016). Indeed, reflecting the results of many studies, Rakovec et al. (2016) showed that streamflow data are necessary but not sufficient to warrant constraining model components by dividing incoming precipitation among storage, evaporation and drainage (Bouaziz et al., 2021). Thus, multiple strategies have been developed to improve the physical realism of conceptual

models (i.e. model *consistency*) (Efstratiadis and Koutsoyiannis, 2010), including using additional data that represent internal hydrological states and flows other than streamflow when estimating parameters. Treating the system more holistically (i.e. forcing models to simulate multiple response variables adequately) has considerable potential to improve model accuracy (Hrachowitz et al., 2014). The value of such multi-variable and/or multi-objective strategies has been demonstrated using groundwater levels (Fenicia et al., 2008; Freer et al., 2004; Giustolisi and Simeone, 2006; Molenat et al., 2005), near-surface soil moisture (Brocca et al., 2010; Kunnath-Poovakka et al., 2016; López López et al., 2017; Rajib et al., 2016; Sutanudjaja et al., 2014), saturated contributing areas (Blazkova et al., 2002; Franks et al., 1998; Güntner et al., 1999), snow cover (Bennett et al., 2019; Gao et al., 2017; Riboust et al., 2019), evaporation (Bouaziz et al., 2018; Demirel et al., 2018; Hulsman et al., 2020), streamflow at subcatchment outlets (Moussa et al., 2007), satellite-based total water storage anomalies (Werth and Güntner, 2010; Yassin et al., 2017) and tracer data (Birkel et al., 2011; Capell et al., 2012; Birkel et al., 2015; Kuppel et al., 2018a; Piovano et al., 2019; Stadnyk and Holmes, 2023). Alternately, one may seek to extract more information from the available data, for example by developing signatures that represent different aspects of the data (Euser et al., 2013; Fenicia et al., 2018; Gharari et al., 2014), and then compare the signatures of the observed and simulated time series. For streamflow, the hydrological signatures can include quantiles of the streamflow distribution (values of the flow duration curve (FDC)), the base flow index, the flashiness index and many others (e.g., Kavetski et al., 2018).

Simultaneously calibrating hydrological models with streamflow and tracer or other solute concentrations in the stream may decrease their uncertainty and increase their physical plausibility because of the need to reproduce both hydrological and biogeochemical dynamics (Birkel et al., 2017; Fovet et al., 2015b; Pesántez et al., 2023; Pettersson et al., 2001; Strohmenger et al., 2021; Woodward et al., 2013a). The value of this strategy has been demonstrated, for example using concentrations of chloride (Hrachowitz et al., 2013) or nitrate ($NO_3^-$) and sulphate (Hartmann et al., 2013; Pettersson et al., 2001). As the movement of water and solutes through the landscape is inherently coupled (Knapp et al., 2020), using time series of multiple elements along with streamflow during calibration may provide additional insights into the flow paths of water through the catchment (Strohmenger et al., 2021). This potential may be particularly high when using solutes that differ in their sources and flow paths across spatial and temporal scales in a catchment. Calibration that includes streamflow along with solutes that have distinct dynamics, as frequently observed with dissolved organic carbon (DOC) and $NO_3^-$ (Inamdar and Mitchell, 2006; Taylor and Townsend, 2010), such as in headwater agricultural catchments (Aubert et al., 2013; Strohmenger et al., 2020; Thomas et al., 2016), thus has high potential to constrain models to adequately reproduce water storage dynamics and flow paths.

The objective of this study was thus to test the hypotheses that, by including daily in-stream DOC and $NO_3^-$ concentrations simultaneously in a parsimonious conceptual model in a multi-objective and multi-variable calibration and evaluation strategy, we could increase the model's (1) ability to predict streamflow for calibration or evaluation periods and (2) internal consistency, and (3) reduce the uncertainty in hydrological parameters.

## 2. Materials and Methods

### 2.1. Study site

The Kervidy-Naizin catchment is located in western France (48° 0' N, 2° 5' W) (Fig. 1) and forms part of the Agro-Hydro Systems (AgrHyS) Critical Zone Observatory (Fovet et al., 2018). It is a 4.82 km² headwater catchment of the 12 km² Naizin catchment (Fig. 1), which is drained by a second-Strahler-order intermittent stream that frequently dries up from July to October. The climate is temperate oceanic, with a mean ± standard deviation of annual temperature of $11 \pm 0.6$°C, annual cumulative precipitation of $894 \pm 170$ mm yr$^{-1}$ and specific discharge of $350 \pm 140$ mm yr$^{-1}$ from 2008-2016. The topography is relatively flat, with few slopes reaching a gradient of 5%, and an elevation range of 98-140 m above sea level. The soil is a silty loam 0.5-1.5 m deep, with well-drained Cambisols in the upslope zone and poorly drained Epistagnic Haplic Luvisols and Albeluvisols in the downslope riparian zone (FAO classification (WRB, 2006)). At the global scale, Kervidy-Naizin is representative of headwater catchments underlain by bedrock in temperate climates. The bedrock consists of impervious, locally fractured Brioverian schists and lies below a fissured and fractured weathered layer of variable thickness 1-30 m deep (Molenat et al., 2005). A shallow, perennial groundwater body develops in the soil and weathered bedrock. In the upland domain, consisting of well-drained soils, the water table remains below the soil surface throughout the year, varying in depth from 1-5 m (Molenat et al., 2005). In the wetland domain, developed near the stream and consisting of hydromorphic soils (hereafter, "riparian zone"), the water table is shallower, remaining near the soil surface generally from October to April/May each year. The seasonal fluctuation of the water table in this catchment has been described as a succession of three hydrological periods (Aubert et al., 2013; Lambert et al., 2013): (i) rewetting of riparian wetland soils after the dry summer season, (ii) rise of groundwater in the upland domain that leads to prolonged waterlogging of wetland soils and establishes a marked hydraulic gradient in groundwater between upland and wetland domains and (iii) drawdown of groundwater that leads to drying of the stream (Humbert et al., 2015).

The land use of Kervidy-Naizin consists mainly of agriculture with intensive mixed crop-livestock farming, with maize (36% of the area), cereals (32%) and grasslands (13%), and a high density of livestock (i.e. dairy cattle, pigs and poultry) of 5 livestock units ha$^{-1}$ (Benoit and Veysset, 2021) according to farm surveys performed in 2008 and 2013 and annual land-use surveys (Casal et al., 2018, 2019; Viaud et al., 2018). From 2002–2015, mean N inputs on the catchment equalled 257 kg ha$^{-1}$ yr$^{-1}$, coming from slurry and manure fertilization (69%), inorganic fertilization (21%, mainly ammonium nitrate), cattle excretion in pastures (5%) and nitrogen (N) fixation (5%) (Casal et al., 2019). Kervidy-Naizin is representative of intensive agricultural areas that have an excess of reactive N due to the application of livestock waste and inorganic fertilisers in excess of crop requirements.

In this landscape, most DOC and $NO_3^-$ accumulate in riparian-zone soils and groundwater, respectively (Aubert et al., 2013; Strohmenger et al., 2020). Using end-member mixing analysis to identify DOC sources and quantify their contributions to the DOC stream in Kervidy-Naizin, Morel et al. (2009) estimated that 64-86% of the DOC that entered the stream during storms, when much of the DOC export from soils to streams and rivers occurs (Lambert et al., 2014), came from riparian wetland soil. This result confirmed previous studies that found that riparian soils are the main source of DOC in most headwater catchments (Lambert et al., 2013). Morel et al. (2009) also demonstrated that this riparian wetland zone in Kervidy-Naizin behaved as non-limiting storage of DOC during flushing. Hillslope soils in this catchment also contribute to stream DOC export, but dissolved organic matter (DOM) in upland soils is supply-limited and seasonally depleted after groundwater rises. Upland DOC contribution decreases from ca. 30% of the stream DOC flow at the beginning of the high-flow period to < 10%

later in this period (Lambert et al., 2013, 2014). In addition, in a high-frequency, multi-solute 10-year monitoring (2000-2010) study of Kervidy-Naizin, Aubert et al. (2013) identified that $NO_3^-$ accumulated in groundwater at a concentration of ca. 20.7 mg N-$NO_3$ $L^{-1}$ compared to 1.6 mg N-$NO_3$ $L^{-1}$ in riparian wetland.

Long-term analysis of the dynamics of nutrient concentrations and hydroclimatic variables at multiple time scales in Kervidy-Naizin highlighted the opposition of dynamics of DOC and $NO_3^-$ concentrations due to opposition in their spatial sources. DOC concentrations peaked under wet or stormflow conditions, when $NO_3^-$ concentrations were lowest. In contrast, $NO_3^-$ concentrations peaked under high-water-table and drier conditions, when DOC concentrations were lowest. This opposition between maxima and minima of daily DOC and $NO_3^-$ concentrations can be interpreted as the result of relative mixing contributions of soil-surface riparian flows (i.e. DOC-rich and $NO_3$-poor) and upslope groundwater flows (i.e. $NO_3$-rich and DOC-poor) (Strohmenger et al., 2020).

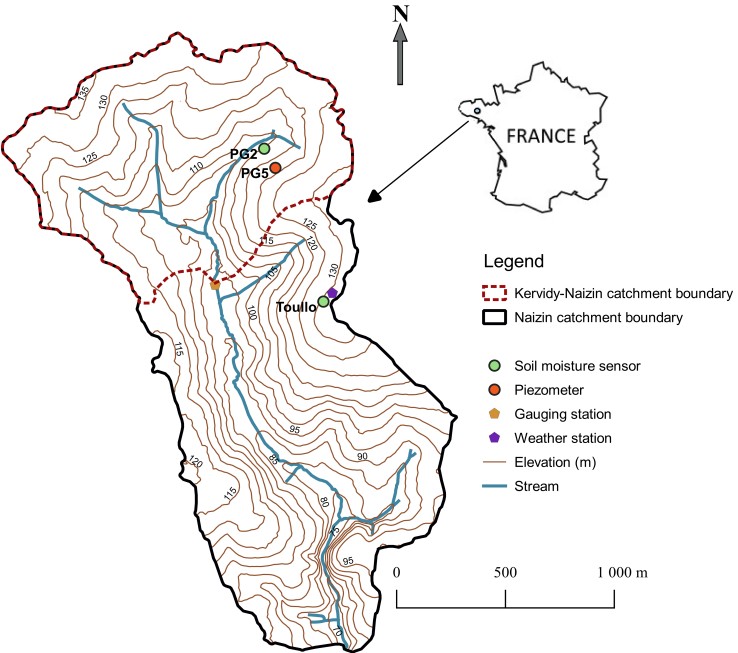

**Figure 1**. Map of the nested Kervidy-Naizin and Naizin catchments (4.82 and 12.00 km², respectively), in western France. Data from the weather station and Toullo station, which lie outside Kervidy-Naizin but inside Naizin, were used in this study.

**2.2. Data monitoring**

We used daily aggregated meteorological and streamflow measurements collected from 2002-2017. The weather station in Kervidy-Naizin (Cimel Enerco 516i), located ca. 1 km from the outlet of the catchment (Fig. 1), records hourly precipitation, air and soil temperatures, air humidity, global radiation, wind direction and wind speed, which allowed for calculation of potential evapotranspiration using the Penman equation (Penman, 1956). Stream level was recorded every minute at the outlet using a float-operated shaft-encoder level sensor and a data logger (Thalimedes OTT) and then converted to streamflow using a rating curve (Carluer, 1998).

Stream water was manually sampled daily at ca. 17:00 at the outlet station. These instantaneous grab samples were immediately filtered (pore size: 0.22 μm) on site and stored in the dark at 4°C in propylene bottles. Analyses were performed within a maximum of two weeks. $NO_3^-$ concentrations were measured by ionic chromatography (DIONEX DX 100, (ISO, 1995), precision: ±2.5%). DOC was estimated as total dissolved carbon (C) minus dissolved inorganic C, both measured using a C analyser (Shimadzu TOC 5050A, precision: ±5%).

Shallow-groundwater data were collected by a piezometer at mid-slope point (PG5, Fig. 1). The groundwater level at PG5, which has been measured every 15 min (Orpheus OTT) since 2000 using pressure probes, was used because its variations are representative of mean variations in the shallow groundwater in Kervidy-Naizin. The volumetric soil water content was measured in upland and riparian zones of the catchment using time domain reflectometry (TDR) probes. In the upland zone (Toullo station, Fig.1), it was measured at three depths (i.e. 5, 20 and 50 cm), with three replicates per depth, at 30 min intervals from 1 Jan 2016 to 1 Jan 2019; these data were first averaged by depth and then aggregated into daily values. Although the Toullo station lies outside Kervidy-Naizin, we assumed that, as Kervidy-Naizin and Naizin are nested, it could represent Kervidy-Naizin's soil moisture conditions in the upland zone. In the riparian zone (point PG2, Fig. 1), the volumetric soil water content was measured at a depth of 5 cm, with three replicates, at 30 min intervals from 3 Dec 2013 to 1 Jan 2017; these data were also averaged and then aggregated into daily values.

### 2.3. Rationale for the solute-transport model

We used a parsimonious semi-distributed solute-transport model, implemented in Python, that was iteratively customized and tested within the DYNAMITE modular modelling framework (Fovet et al., 2015a; Hrachowitz et al., 2014, 2021). The processes are represented by linear or non-linear equations that connect the flows to model reservoirs (Beven, 2012). This representation of storage-discharge relationships directly connects water flows to biogeochemical processes, which facilitates simultaneous simulation of both water and solute flows (Birkel et al., 2017).

### 2.3.1. Hydrology

The model spatially distinguishes two functionally distinct response units: hillslope and riparian zones. It represents them as two parallel suites of reservoirs connected by a common groundwater reservoir (Fig. 2). The hillslopes are represented as two reservoirs: the rooting-zone reservoir ($S_U$) [L] and a fast-responding reservoir ($S_F$) [L] (e.g. preferential flow structures). As riparian zones often have a distinct hydrological function (Molenat et al., 2005; Seibert et al., 2003, 2009), the model also represents them as two reservoirs: an unsaturated-zone reservoir ($S_{UR}$) [L] and a fast-responding reservoir ($S_R$) [L]. The two parallel suites are connected by a slow groundwater reservoir ($S_S$) [L], characterized by a threshold from which the groundwater feeds the $S_{UR}$ reservoir that represents a groundwater mixing volume ($S_{S\_mix}$) [L]. See Table 1 for the relevant model equations. More detailed model description and justifications for the processes modelled can be found in previous studies (Hrachowitz et al., 2013, 2014, 2015).

The rainfall-runoff model uses daily precipitation (P) [L T$^{-1}$] and potential evapotranspiration ($E_P$) [L T$^{-1}$] to simulate daily specific discharge at the outlet ($Q_T$) [L T$^{-1}$]. Upon reaching the soil, P is divided into water that infiltrates into $S_U$ ($R_U$, Table 1) and excess water by a hillslope runoff-generation coefficient ($C_{H,R}$) routed to $S_F$ ($R_F$) and $S_S$ ($R_P$). $C_{H,R}$ is estimated by a logistic function representing the catchment-wide soil water holding capacity in the rooting zone ($S_{U\_max}$), which roughly reflects soil water content at field capacity, and a shape factor ($\beta_H$). Percolation of water from $S_U$ to $S_S$ ($R_{SS}$) is estimated by a linear function of the water storage in $S_U$ and a maximum percolation capacity ($P_{max}$). Evapotranspiration from $S_U$ ($E_U$) is estimated by a linear function of the relative soil moisture and a transpiration threshold ($L_P$), which is the fraction of $S_{U\_max}$ below which potential evapotranspiration ($E_P$) is constrained by the water available in $S_U$.

Fast reservoir $S_F$ receives water ($R_F$) from $S_U$ (Table 1, Eq. (8)) and drains into reservoir $S_{UR}$ according to a linear storage-discharge relationship that is controlled by parameter $k_F$. Slow reservoir $S_S$ is recharged by $R_{SS}$ and $R_P$

from $S_U$ and slowly drains according to a linear storage-discharge relationship that is controlled by parameter $k_S$. The water drained from $S_S$ is redistributed between $S_{UR}$ and the stream according to parameter $f_{SUR}$. Deep-infiltration losses from $S_S$, represented by calibration parameter $Q_L$, are used to explicitly represent inter-catchment groundwater flows (i.e. groundwater flows that cross topographic divides), implying that precipitation that falls in one catchment influences the streamflow in another catchment (Bouaziz et al., 2018). Analysis of the long-term water balance of a headwater catchment with similar physiography in Brittany revealed a large deficit (Hrachowitz et al., 2014). There is evidence that many catchments have such deficits, which are caused, at least in part, by large inter-catchment groundwater flow (Hrachowitz et al., 2014; Le Moine et al., 2007), although this cannot be verified completely, as highlighted by Beven (2001). In addition, data from 58 catchments in the Meuse basin indicated that large net inter-catchment groundwater flows likely existed, mainly in small headwater catchments underlain by fractured aquifers (Bouaziz et al., 2018), such as Kervidy-Naizin. The parameter for deep-infiltration losses is also used to reproduce the zero flow at the outlet and groundwater dynamics with a long recession observed during the summer, regardless of the piezometer (Humbert et al., 2015). Consequently, we explicitly modelled inter-catchment groundwater flows for Kervidy-Naizin. Common conceptual models rarely include deep-infiltration losses, which may not prevent them from simulating streamflow accurately, but may cause them to misrepresent the natural system, particularly by overestimating actual evaporation rates in compensation (Bouaziz et al., 2018). In the present study, in the absence of detailed knowledge of the underlying processes, deep-infiltration losses from Kervidy-Naizin were conceptualized as a loss term $Q_L$ from $S_S$.

Riparian reservoir $S_{UR}$ receives water from $S_F$, $S_S$ and precipitation (Table 1, Eq. (13)). Excess water, estimated using a runoff-generation coefficient ($C_{R,R}$), is routed to $S_R$ ($R_R$). The water that remains in $S_{UR}$ is available for transpiration ($E_{UR}$, Table 1, Eq. (14)). $S_R$ drains into the stream according to a linear storage-discharge relationship that is controlled by parameter $k_R$ (Table 1, Eq. (18)). The total simulated stream discharge equals the sum of slow and fast contributions from $S_S$ and $S_R$, respectively (Table 1, Eq. (19)).

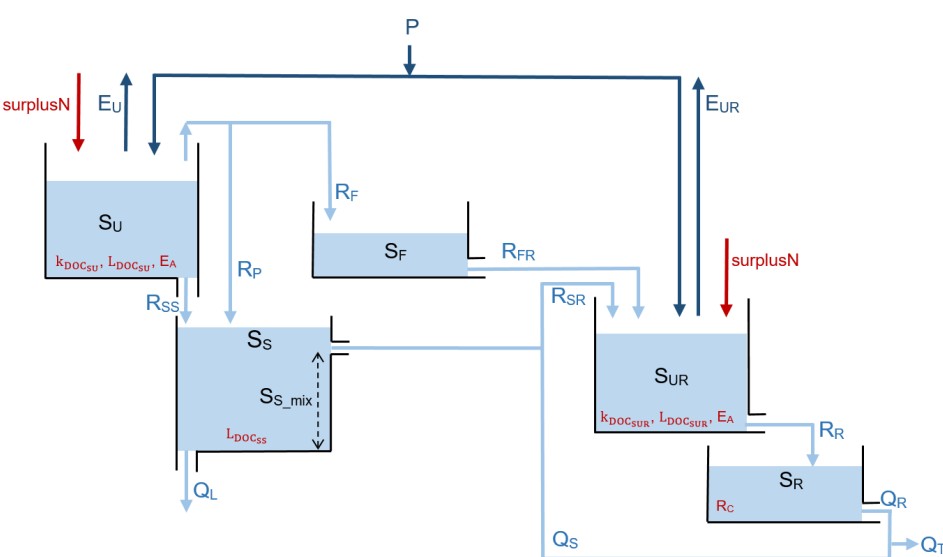

**Figure 2**. Conceptual model structure used to represent the Kervidy-Naizin catchment. S are storage components, R are recharge flows between reservoirs, Q are liquid flows that leave the system and E are evaporative flows that leave the system. Dark blue and light blue arrows represent water flows and water with solutes, respectively. Biochemical parameters are shown in red for each reservoir. See Table 2 for definitions of the parameters and Table A1 for definitions of the variable abbreviations.

**Table 1.** State and flow equations of the model. See Table A1 for definitions of the variable abbreviations.

| Process | Water balance | Eq. | Flow and state equations, and relationships | Eq. |
|---|---|---|---|---|
| Unsaturated zone | $dS_U/dt = P - E_U - R_F - R_P - R_{SS}$ | (1) | $E_U = E_P \cdot min\left(1, \dfrac{S_U}{S_{U\_max}} \dfrac{1}{L_P}\right)$ | (2) |
| | | | $R_U = (1 - C_{H,R}) \cdot P$ | (3) |
| | | | $R_F = C_{H,R} \cdot (1 - C_P) \cdot P$ | (4) |
| | | | $R_P = C_{H,R} \cdot C_P \cdot P$ | (5) |
| | | | $R_{SS} = P_{max} \cdot \left(\dfrac{S_U}{S_{U\_max}}\right)$ | (6) |
| | | | $C_{H,R} = \dfrac{1}{\left(1 + exp\left(\dfrac{-S_U/S_{U\_max} + 0.5}{\beta_H}\right)\right)}$ | (7) |
| Fast reservoir | $dS_F/dt = R_F - R_{FR}$ | (8) | $R_{FR} = S_F \cdot (1 - e^{-k_F t}) dt^{-1}$ | (9) |
| Slow reservoir | $dS_S/dt = (1 - f) \cdot (R_{SS} + R_P) - Q_S - R_{SR} - Q_L$ | (10) | $Q_S = \begin{cases} (S_S - S_{S\_mix} - Q_L) \cdot (1 - f_{SUR}) \cdot (1 - e^{-k_S t}) dt^{-1}, & (S_S - S_{S\_mix} - Q_L) > 0 \\ 0, & (S_S - S_{S\_mix} - Q_L) \leq 0 \end{cases}$ | (11) |
| | | | $R_{SR} = \begin{cases} (S_S - S_{S\_mix} - Q_L) \cdot f_{SUR} \cdot (1 - e^{-k_S t}) dt^{-1}, & (S_S - S_{S\_mix} - Q_L) > 0 \\ 0, & (S_S - S_{S\_mix} - Q_L) \leq 0 \end{cases}$ | (12) |
| Riparian unsaturated reservoir | $dS_{UR}/dt = P + \dfrac{R_{FR} \cdot (1 - f)}{f} + \dfrac{R_{SR}}{f} - E_{UR} - R_R$ | (13) | $E_{UR} = E_P \cdot min\left(1, \dfrac{S_{UR}}{S_{UR\_max}} \dfrac{1}{L_P}\right)$ | (14) |
| | | | $R_R = C_{R,R} \cdot P$ | (15) |
| | | | $C_{R,R} = min\left(1, \left(\dfrac{S_{UR}}{S_{UR\_max}}\right)^{\beta_R}\right)$ | (16) |
| Riparian reservoir | $dS_R/dt = R_R - Q_R$ | (17) | $Q_R = S_R \cdot (1 - e^{-k_R t}) dt^{-1}$ | (18) |
| Total runoff | $Q_T = Q_S + f \cdot Q_R$ | (19) | | |
| Total evaporative flows | $E_A = (1 - f) \cdot E_U + f \cdot E_{UR}$ | (20) | | |

### 2.3.2. Nitrate transfer and transformation

N inputs to reservoirs $S_U$ and $S_{UR}$ are the daily N surplus (kg N ha$^{-1}$), which correspond to soil N balances. N inputs consist of inorganic and organic fertilisers (i.e. slurry and manure), biological N fixation and atmospheric N deposition. N outputs equal the sum of N exported by each crop type. In this study, the N surplus was considered as a net (N inputs - N outputs) diffuse N source for the catchment (Dupas et al., 2020). Farm surveys performed in 2008 and 2013 led to estimates of a mean annual surplus over the study period (2002–2017) of ca. 90 kg N ha$^{-1}$ y$^{-1}$ (Casal, 2018). Given the uncertainty in the estimated N surplus, we considered it as a calibration parameter (surplusN, Table 2).

Due to the lack of relevant studies, the period with the highest heterotrophic denitrification rate is unknown for Kervidy-Naizin. In agricultural headwaters, denitrification rates are usually low at the end of winter, increase in spring, peak in summer and decrease in autumn before reaching their lowest in the middle of winter (Anderson et al., 2014). In agricultural landscapes where N availability exceeds plant requirements, denitrification is limited mainly by C availability, O$_2$ concentration and temperature (Barton et al., 1999). Riparian zones of these landscapes often contain large amounts of C. Thus, denitrification rates are expected to be highest from late spring to early autumn, when temperatures are highest and, as long as soils remain wet, O$_2$ concentrations are lowest (Anderson et al., 2014). We also had no observations of biological transformation of $NO_3^-$ through consumption by aquatic primary producers, although we assumed that it was highest in spring and summer. Thus, in the absence of detailed knowledge of the temporal pattern of biological $NO_3^-$ removal in Kervidy-Naizin, we represented biological transformation of $NO_3^-$ as a constant annual amount of $NO_3^-$ removal (Rc) (kg N ha$^{-1}$ yr$^{-1}$) from reservoir

S$_R$ (Table 2). We assumed that if this constant overestimated the biological $NO_3^-$ removal usually observed in agricultural landscapes in winter, it would influence $NO_3^-$ concentration little given the Kervidy-Naizin's high $NO_3^-$ load in winter. Thus, representing biological removal as a constant was assumed to be reasonable in a parsimonious model approach (Fovet et al., 2015b).

Denitrification can be a sink for $NO_3^-$ in streams, particularly small (low-order) ones (Böhlke et al., 2009). However, methods for measuring in-stream denitrification are difficult and have high uncertainty, and the controlling variables are not known well enough to make reliable predictions for targeted management decisions (Böhlke et al., 2009). Given the lack of in-stream denitrification observations and the low potential for in-stream $NO_3^-$ removal (estimated at ca. 4% per year; Salmon-Monviola et al. (2013)) in Kervidy-Naizin, we did not model it and thus assumed zero in-stream denitrification.

**2.3.3. Dissolved organic carbon transfer and transformation**

The conceptualization of biogeochemical processes used to simulate DOC dynamics, similar to that of Birkel et al. (2014), is based on a simple production-loss mass balance and transport along the main flow pathways to the stream. The DOC mass balance ($\Delta mass_{DOC_i}$ [M]) during time step $\Delta t$ [T] ($\Delta t$ = 1 day, in this study) of each reservoir $i$ (i.e. S$_U$, S$_{UR}$ and S$_S$) differs from more complex C-process models by being simplified into a grouped representation of DOC production ($Production_{DOC_i}$ [M]) (processes that transform C were not distinguished) and loss ($Loss_{DOC_i}$ [M]) (processes that consume, retain and mineralize DOC were not distinguished) (Di Grazia et al., 2023; Koch et al., 2013) :

$$\Delta mass_{DOC_i} = Production_{DOC_i} - Loss_{DOC_i} \tag{21}$$

DOC production ($Production_{DOC_i}$ [M]) of reservoir $i$ are calculated by multiplying DOC concentration ($[DOC]_i$ [M L$^{-1}$]) with the total water stored ($S_i$ [L]) at the beginning of each time step. DOC production was assumed to increase as temperature and soil water content increased (Birkel et al., 2020) :

$$[DOC]_i = k_{DOC_i} \cdot \frac{S_i}{S_{i\_max}} \cdot E_a^{(T-\overline{T})} \tag{22}$$

where $k_{DOC_i}$ [M L$^{-1}$] is the concentration at which DOC is produced daily in a reservoir $i$, E$_A$ (dimensionless) is a calibrated temperature-dependent activation energy, T [°C] is the observed daily air temperature, $\overline{T}$ [°C] is the mean annual air temperature for the study period, $S_{i\_max}$ and $S_i$ the capacity [L] and total water stored [L], respectively, of reservoir $i$. DOC was assumed not to be produced in the groundwater reservoir (S$_S$), as deeper mineral horizons in soil are considered to be DOC sinks instead (Kalbitz and Kaiser, 2008) and low DOC concentrations have been observed in Kervidy-Naizin's groundwater (mean of ca. 1 mg L$^{-1}$; Aubert et al. (2013)). However, DOC can accumulate in S$_S$ due to recharge from the hillslope reservoir (S$_U$).

Potential DOC losses ($Loss_{DOC_i}$ [M]) in the form of mineralization (Köhler et al., 2002), absorption or consumption in reservoirs S$_U$, S$_{UR}$ and S$_S$ are calculated using a loss coefficient ($L_{DOC_i}$) (dimensionless) (Table 2) applied to the DOC mass of reservoirs at the beginning of each time step.

We assumed that in-stream processes have negligible influence on DOC concentrations. Some studies found that agricultural land use can increase the production of autochthonous DOM in streams (Shang et al., 2018). For example, in an agricultural catchment (Lower Austria, 66 ha), one large DOC source was the stream itself, as in-stream processes caused 37% of the total DOC load measured at the catchment outlet during base flow conditions from November to May (Eder et al., 2022). Nevertheless, end-member mixing analysis of DOC in Kervidy-Naizin found that stream DOC dynamics during winter storm events could be explained by catchment processes, with

little contribution from in-stream sources (Morel et al., 2009). These results confirmed that most of the DOC in streams that drain headwater catchments is likely to be of external origin (i.e. allochthonous), resulting from interactions between biogeochemical and hydrological processes in soils, at least during the wet season (Dalzell et al., 2007; Fovet et al., 2020; Lambert et al., 2013, 2014; Raymond and Saiers, 2010). This is also consistent with the theory of DOM transformation along a fluvial continuum (Creed et al., 2015) and the dynamics of DOM fluorescence observed for example by Shang et al. (2018), who found increasing contribution of protein-like autochthonous DOM, accompanied by decreasing contribution of allochthonous DOM, from low-order to high-order systems. For Kervidy-Naizin, these results are supported by two arguments. First, some processes associated with DOC production in summer are unlikely to occur in Kervidy-Naizin's stream, which frequently dries up from July to October. Second, riparian vegetation is dense and covers the entire length of Kervidy-Naizin's network, which decreases primary production of DOC. Thus, we considered the assumption regarding the negligible influence of in-stream processes on DOC concentrations to be valid for Kervidy-Naizin.

The daily solute ($NO_3^-$ or DOC) concentration at the outlet ($C_{out_{solute}}$ [M L$^{-1}$]) is then calculated according to the relative contribution of reservoirs $S_S$ and $S_R$:

$$C_{out_{solute}} = \frac{C_{solute_{S_S}} \cdot Q_S + C_{solute_{S_R}} \cdot Q_R}{Q_T} \tag{23}$$

**Table 2**. Definitions and uniform prior distributions of the parameters of the solute-transport model.

| Module | Parameter | Unit | Initial Range | Definition |
|---|---|---|---|---|
| **Rainfall-Runoff** | $S_{U\_max}$ | [mm] | [50-1000] | Storage capacity of the hillslope unsaturated zone |
| | $C_P$ | [-] | [0.005-1.0] | Preferential recharge coefficient |
| | $\beta_H$ | [-] | [0.01-4] | Hillslope runoff coefficient |
| | $P_{max}$ | [mm d$^{-1}$] | [0.1-6] | Percolation capacity |
| | $L_P$ | [-] | [0.01-0.8] | Transpiration threshold |
| | $k_F$ | [d$^{-1}$] | [0.001-1] | Storage coefficient of the fast reservoir |
| | $k_S$ | [d$^{-1}$] | [0.02-0.06] | Storage coefficient of the slow reservoir |
| | $S_{S\_mix}$ | [mm] | [500-9000] | Groundwater mixing volume in the slow reservoir |
| | $f_{SUR}$ | [-] | [0.00001-0.2] | Proportion of water flow from reservoir S$_S$ that passes through reservoir S$_{UR}$ |
| | $Q_L$ | [mm d$^{-1}$] | [0.05-1] | Deep infiltration loss |
| | $f$ | [-] | [0.15-0.30] | Proportion of the catchment covered by the riparian zone |
| | $S_{UR\_max}$ | [mm] | [50,500] | Storage capacity in the riparian unsaturated zone |
| | $\beta_R$ | [-] | [1-7] | Riparian runoff coefficient |
| | $k_R$ | [d$^{-1}$] | [0.04-2] | Storage coefficient of the riparian reservoir |
| **Nitrate** | surplusN | [kg N ha$^{-1}$ year$^{-1}$] | [50-95] | Nitrogen surplus |
| | Rc | [kg N ha$^{-1}$ year$^{-1}$] | [25-40] | Amount of nitrate removed |
| **Dissolved organic carbon (DOC)** | $k_{DOC_{SU}}$ | mg L$^{-1}$ | [15-35] | DOC concentration in unsaturated storage |
| | $k_{DOC_{SUR}}$ | mg L$^{-1}$ | [15-35] | DOC concentration in riparian storage |
| | $E_A$ | [-] | [1.0-1.2] | Energy parameter |
| | $L_{DOC_{SU}}$ | [-] | [0-1] | DOC loss in unsaturated storage |
| | $L_{DOC_{SS}}$ | [-] | [0-1] | DOC loss in slow storage |
| | $L_{DOC_{SUR}}$ | [-] | [0-1] | DOC loss in riparian storage |

### 2.3.4. Mixing assumption

Each reservoir in the model is assumed to be completely mixed to simulate solute dynamics. This approach, used in most studies based on conceptual models (Birkel et al., 2020; McMillan et al., 2012; Pesántez et al., 2023), assumes instantaneous and complete mixing of the incoming water and solute masses in each reservoir, according to a solute-balance equation:

$$\frac{d(c_i \cdot S_i)}{dt} = \sum_j c_{I,j} \cdot I_j - \sum_k c_{O,k} \cdot O_k \tag{24}$$

where $S_i$ is the amount of water stored in reservoir $i$ [L], $c_i$ is the associated solute concentration [M L$^{-1}$], I are the j water-inflow [L T$^{-1}$] to a given reservoir (e.g. $R_{SS}$ and $R_P$ from $S_U$ to $S_S$) (Fig. 2) with the corresponding solute-inflow concentrations $c_{I,j}$ [M L$^{-1}$], and O are the k water-outflow [L T$^{-1}$] from a given reservoir with the corresponding solute-outflow concentrations $c_{O,k}$ [M L$^{-1}$] (e.g. $R_{SR}$ and $Q_S$ from $S_S$) (Fig. 2).

The model tracks the distribution of ages of the water outflow ($p_{Outflow}(T, t)$, where T is the transit time at time $t$) (Benettin et al., 2022) using a time stamp for each daily incoming and outflowing water flow in reservoirs, similar to the approach of Birkel and Soulsby (2016). The distribution of ages of water in a reservoir ($p_S(T, t)$) can be derived in a similar way to tracking the ages of water in outflow ($p_{Outflow}(T, t)$), as they are related by a StorAge-Selection (SAS) function developed by Botter et al. (2011):

$$\omega_{Outflow}(T,t) = \frac{p_{Outflow}(T,t)}{p_S(T,t)} \tag{25}$$

The SAS function can be considered a statistical summary of the transport behaviour of a hydrological system that quantifies the release of water of different ages from a reservoir to an outflow (Rinaldo et al., 2015). According to the complete mixing assumption of the model, the age distributions of storage and flow are identical to each other (i.e. the outflow composition perfectly represents the storage composition) (Benettin et al., 2022). Thus, the solute concentration of outflow equals the solute concentration of the reservoir. This "well-mixed" situation corresponds to uniform sampling in which $\omega_{Outflow}(T,t) = 1$ and implies that water storage is uniformly sampled by an outflow (Benettin et al., 2013).

### 2.4. Sensitivity analysis of the solute-transport model

A global sensitivity analysis (GSA) was carried out to determine the effect of the model calibration scenarios on the most sensitive hydrological parameters. GSA allows to identify the extent to which changes in different parameters influence changes in the hydrological model output, and to determine the most important parameters (i.e. that need to be calibrated) and the least important parameters (i.e. that can be fixed as constants) (Reusser et al., 2011; Wang and Solomatine, 2019). GSA, which ranks the relative influence of model parameters on model output (Sun et al., 2022), is generally recommended for hydrological models due to its advantages over local sensitivity analysis methods. Indeed, GSA can consider the influence of input parameters over their entire range of variation and is suitable for non-linear and non-monotonic models, providing results that are independent of modeller bias and a particular site (Song et al., 2015). Among the GSA methods widely applied to hydrological models, we chose a variance-based method as it can provide the most accurate and robust sensitivity indices for complex non-linear models (Reusser et al., 2011; Song et al., 2015; Wang and Solomatine, 2019). Variance-based methods assume that a parameter's influence can be measured by the contribution of the parameter itself or its interactions with two or more other parameters to the variance of the output. The main advantage of variance-based methods is that they can calculate the main and higher-order effects of parameters, which identifies which ones strongly influence the output on their own, and which ones strongly influence the output due to their

interactions with other parameters (Wang and Solomatine, 2019). We used the Fourier Amplitude Sensitivity Test (FAST) (Saltelli et al., 1999) from the SPOTPY Python framework (Houska et al., 2015) to calculate variance-based sensitivity indices that ranged from 0-1. FAST calculates a first-order sensitivity index ($S_i$), which measures the effect of each parameter on the output, and a total sensitivity index ($S_{Ti}$), which measures the effect of each parameter and its interactions with the other parameters on the output (Shin and Kim, 2017). Because $S_{Ti}$ provides more reliable results than $S_i$ when investigating the overall influence of each parameter on the output (Saltelli et al., 2009), we used it to investigate parameter sensitivity, as defined by Saltelli and Annoni (2010):

$$S_{Ti} = \frac{E_{X_{\sim i}}\left(V_{X_i}(Y|X_{\sim i})\right)}{V(Y)} \tag{26}$$

where $X_i$ is the $i^{th}$ parameter, and $X_{\sim i}$ is the vector of all parameters except $X_i$.

The variance between parentheses in the numerator denotes that the variance of Y, the value of the scalar objective function, is considered over all possible values of $X_i$ while keeping $X_{\sim i}$ fixed. The expectation operator outside the parentheses is considered over all possible values of $X_{\sim i}$, while the variance V(Y) in the denominator is the total (unconditioned) variance (Shin and Kim, 2017). The numerator represents the expected variance if all parameters except $X_i$ are fixed (Saltelli and Annoni, 2010).

Calculating $S_{Ti}$ for a single parameter requires n×(p+2) model runs, where n is the sample size and p is the number of parameters (Saltelli, 2002). To determine an appropriate sample size for this GSA, we relied on the experiment of Nossent et al. (2011), in which the sensitivity index did not converge until n = 12,000; thus, with 14 hydrological parameters, we performed 192,000 model runs. In this GSA, the Nash-Sutcliffe model efficiency coefficient (Nash and Sutcliffe, 1970) was used to assess daily streamflow output, as suggested by Nossent et al. (2011).

## 2.5. Model calibration and evaluation

To limit adverse effects of equifinality and ensure robust posterior parameter distributions to represent processes meaningfully, extensive multi-objective and multi-variable calibration was performed by calibrating hydrological and biogeochemical model predictions simultaneously. When using multi-objective optimization to calibrate a model, the goal is to find a set of solutions that simultaneously optimize several, potentially conflicting, objective functions that measure individual processes. The interaction of multiple objectives leads to a set of compromised solutions known as Pareto-optimal front (Mostafaie et al., 2018). As none of the solutions can be considered superior when there is more than one objective to optimize, Pareto-optimal solutions (hereafter, "Pareto front") are also called non-dominated solutions (Yeste et al., 2023) with equally good parameter sets, which provides an uncertainty boundary of the predictive model. The caRamel algorithm (Monteil et al., 2020) used in this approach combines the multi-objective evolutionary annealing-simplex algorithm (Efstratiadis and Koutsoyiannis, 2008) and the non-dominated sorting genetic algorithm II (Reed and Devireddy, 2004). The caRamel algorithm produces an ensemble of parameter sets (i.e. a "generation") to run the model, downscales the generation to the parameter sets that optimize the objective functions and generates a new parameter set that produces more accurate results.

The research hypotheses of this study were tested using a stepwise strategy with four model-calibration scenarios based on different combinations of model-performance metrics (Table 3):

- Scenario 1 (S1): only data on streamflow used for calibration, with six metrics used to describe the predicted streamflow signatures
- Scenario 2 (S2): data on streamflow and stream DOC concentration used for calibration, with two metrics including the mean of the metrics in S1 and the Kling–Gupta efficiency (Gupta et al., 2009) used to assess the predicted DOC concentrations

- Scenario 3 (S3): same as S2, but the solute was $NO_3^-$ instead of DOC

- Scenario 4 (S4): data on streamflow and stream DOC and $NO_3^-$ concentrations used for calibration, with three metrics including the mean of the metrics in S1 and the Kling–Gupta efficiency used to assess the predicted DOC and $NO_3^-$ concentrations

The calibration period was set from 1 Jan 2013 to 1 Sep 2016, while the evaluation period was set from 1 Aug 2008 to 31 Dec 2011, each simulated after 3 years of initialization. These periods, the same as those of Strohmenger
et al. (2021), were chosen to be able to compare model performance to two approaches to solute modelling. The hydrological year 2012 was excluded from these periods due to a problem with laboratory analysis of $NO_3^-$ concentrations that year. The uniform prior parameter distributions were based on previous studies of headwater catchments in similar physiographic contexts (Fovet et al., 2015a; Hrachowitz et al., 2015) (Table 2). The prior distribution of storage coefficient $k_S$ had been narrowly constrained based on previous baseflow-recession analysis
using a correlation method (Yang et al., 2018). Three prior parameter constraints (Gharari et al., 2014; Hrachowitz et al., 2014) were added to the calibration algorithm to reduce parameter uncertainties: $k_S < k_F$, $k_F < k_R$ and $S_{UR\_max} < S_{U\_max}$.

Up to 70,000 model runs were used for each calibration scenario, with several successive optimizations to confirm reproducibility of the results, as recommended by Monteil et al. (2020). All parameter sets that belonged to the
final Pareto fronts (hereafter, "envelope") were retained as feasible solutions for each calibration scenario (Table 3). To illustrate the results for the predicted discharges and solute concentrations, a "best-compromise" set was selected from the Pareto front that minimized the Euclidean distance to the optimal point in the multi-objective space of each calibration scenario. All simulated discharges and concentrations using all parameter sets of the Pareto front provided information about the uncertainty in the model's output.

In the later evaluation step, observed soil water content and groundwater level measurements were used as independent data to assess the consistency of internal processes of the best-compromise model for each scenario. Soil moisture is a key variable for the energy and water balance at the land surface. It affects the partitioning of solar radiation into latent and sensible heat as well as the partitioning of precipitation into direct runoff and catchment storage (Duethmann et al., 2022). Accurate prediction of soil moisture is thus essential for simulating
streamflow, evapotranspiration and percolation (Rajat and Athira, 2021; Rajib et al., 2016) and for constraining the parameters of hydrological models. The role of groundwater in the seasonal and multi-year dynamics of streamflow is also essential: in many temperate catchments, groundwater stores water during wet periods and releases it throughout the year, thus contributing greatly to low flows (Pelletier and Andréassian, 2022). These variables are important for characterizing the internal hydrological dynamics of a catchment and are therefore
relevant for assessing the internal consistency of the model.

The data observed for soil water content at Toullo and PG2 were normalized (from 0-1) as a function of their minimum and maximum values over all of the periods studied. All normalized data observed at Toullo station and point PG2 were compared to the normalized simulated water content in the hillslope reservoir ($S_U$) and riparian reservoir ($S_{UR}$), respectively. To compare to the observed groundwater level, the simulated groundwater level was
estimated from simulated water storage in the groundwater reservoir ($S_S$) (Seibert, 2000) using the exponential function $z = -e^{A*S_S+B}$, where $S_S$ is water storage in the slow reservoir, and $z$ is the groundwater level. Coefficients A and B were determined by linear regression between the simulated water storage and the observed groundwater level. The non-parametric Mann-Whitney $U$ test was used to test whether model predictions of calibration scenarios S2, S3 and S4 differed significantly ($p < 0.05$) from those of the baseline scenario S1.


**Table 3**. Signatures for streamflow, dissolved organic carbon (DOC) and nitrate ($NO_3^-$) and the associated performance metrics used for model calibration scenarios and evaluation. The size of the Pareto front was the number of solutions. NSE: Nash–Sutcliffe model efficiency coefficient, KGE: Kling–Gupta efficiency. See Appendix C for definitions of the signatures and performance metrics.

| Calibration scenario | Variables/Signatures | Abbreviation | Performance metrics | Size of the Pareto front | References |
|---|---|---|---|---|---|
| S1: streamflow only | Time series of streamflow | Q | $NSE_Q$, $KGE_Q$ | | Nash and Sutcliffe,1970; Gupta et al., 2009 |
| | Log(Q) | Log(Q) | $NSE_{logQ}$ | | |
| | Flow duration curve | FDC | $NSE_{FDC}$ | 280 | Jothityangkoon et al., 2001 |
| | Runoff ratio | RUNOFF | $NSE_{RUNOFF}$ | | Yadav et al., 2007 |
| | Volumetric efficiency | VE | $VE_Q$ | | Criss and Winston, 2008 |
| S2: streamflow and DOC | Streamflow | Q | Mean of metrics of S1 | 180 | |
| | DOC | DOC | $KGE_{DOC}$ | | Gupta et al., 2009 |
| S3: streamflow and $NO_3^-$ | Streamflow | Q | Mean of metrics of S1 | 110 | |
| | $NO_3^-$ | $NO_3^-$ | $KGE_{NO3}$ | | Gupta et al., 2009 |
| S4: streamflow, DOC and $NO_3^-$ | Streamflow | Q | Mean of metrics of S1 | | |
| | DOC | DOC | $KGE_{DOC}$ | 270 | Gupta et al., 2009 |
| | $NO_3^-$ | $NO_3^-$ | $KGE_{NO3}$ | | Gupta et al., 2009 |


## 3. Results

### 3.1. Global sensitivity analysis of parameter influence on streamflow

The hydrological parameters that influenced predicted streamflow the most were related to recharge ($C_P$; $S_T$ = 0.59), deep-infiltration losses ($Q_L$; $S_T$ = 0.25), percolation capacity ($P_{max}$; $S_T$ = 0.18), storage capacity of the hillslope unsaturated zone ($S_{U\_max}$; $S_T$ = 0.15) and storage coefficient of the fast-responding reservoir in riparian zone reservoir ($k_R$; $S_T$ = 0.14) (Fig. 3). The strong influence of $C_P$ was logical, as it determines the recharge from $S_U$ to $S_S$ and $S_{UR}$ to $S_R$ (i.e., how water from runoff is redistributed between the riparian zone and groundwater). Parameters related to the area of the riparian zone (f) and the transpiration threshold ($L_P$) had less influence.

### 3.2. Prediction of streamflow and solute concentrations

Overall, the model reproduced the main features of the observed hydrological response (Fig. 4) in both the calibration ($NSE_Q$, $NSE_{logQ}$ and $KGE_Q$ > 0.8) and evaluation ($NSE_Q$, $NSE_{logQ}$ and $KGE_Q$ > 0.7) periods for all scenarios. The predicted streamflow reproduced the seasonal dynamics observed during the wetting-up (rising limb of the hydrograph), wet and recession periods. The high flow variations associated with storm events were usually represented relatively well ($NSE_Q$ > 0.75) in calibration and evaluation periods, with good synchronicity, particularly in winter 2010 and 2014. Overall, model performances for the evaluation period were only slightly lower than those for the calibration period for all four scenarios (Figs. 4 and A1). Performance of the best-compromise model was slightly higher for S1 than for the other scenarios, for both calibration and evaluation periods (e.g. comparing S1 ($NSE_Q$ = 0.91, $NSE_{logQ}$ = 0.95, $KGE_Q$ = 0.92) to S4 ($NSE_Q$ = 0.87, $NSE_{logQ}$ = 0.92, $KGE_Q$ = 0.84) for the calibration period) (Fig. 4). The difference in performance between S1 and S2 was smaller. The uncertainty in predicted streamflow estimated from the envelope was low for the calibration and evaluation periods, but appeared to peak during low flow periods. The calibrated model provided similarly reasonable representations of DOC (Fig. 5) and $NO_3^-$ (Fig. 6) concentrations. Predicted DOC concentrations for the calibration period were slightly more accurate for S2 (Fig. 5a) (i.e. $KGE_{DOC}$ = 0.78, $RMSE_{DOC}$= 2.14 mg L$^{-1}$) than for S4 (Fig.

5b) (i.e. $KGE_{DOC} = 0.76$, $RMSE_{DOC} = 2.28$ mg L$^{-1}$). Predicted $NO_3^-$ concentrations for the calibration period were slightly more accurate for S3 (Fig. 6a) (i.e. $KGE_{NO3} = 0.76$, $RMSE_{NO3} = 1.87$ mg N-NO$_3$ L$^{-1}$) than for S4 (Fig. 6b) (i.e. $KGE_{NO3} = 0.74$, $RMSE_{NO3} = 1.95$ mg N-NO$_3$ L$^{-1}$). The model reproduced the contrasting dynamics of stream DOC and $NO_3^-$ (Aubert et al., 2013; Strohmenger et al., 2020), with maximum DOC and minimum $NO_3^-$ concentrations occurring in autumn. During this period, the median simulated DOC concentration was ca. 8.7 mg L$^{-1}$, while that of $NO_3^-$ concentration was ca. 11 mg N-NO$_3$ L$^{-1}$. During the wetting-up period, DOC concentrations decreased to a median of 2.5-3.5 mg L$^{-1}$, while $NO_3^-$ concentrations increased to a median of 14-16 mg N-NO$_3$ L$^{-1}$. These concentrations remained relatively stable during the wet and recession periods. At the end of the recession period, DOC concentration increased slightly to a median of ca. 5.5-6 mg L$^{-1}$, while $NO_3^-$ concentration decreased to a median of ca. 12 mg N-NO$_3$ L$^{-1}$. The model simulated high $NO_3^-$ concentrations in summer, when streamflow and $NO_3^-$ concentrations had not been observed. During summer dry periods, the stream effectively dries up and no water flows at the outlet, which made it more difficult to calibrate the model to predict their solute concentrations. The model simulated near-zero water flow during dry periods, but occasionally simulated flow on certain days when zero flow had been observed, which yielded relatively high simulated $NO_3^-$ concentrations. The lack of observed $NO_3^-$ concentrations during dry periods also provided no constraints that could help the model represent $NO_3^-$ concentrations realistically.

The simulated hydrological signatures for all solutions on the Pareto front provide evidence that including solute data in the calibration improves the ability of the model to reproduce certain streamflow characteristics. While the performance based on median hydrological metrics ($NSE_Q$, $NSE_{logQ}$, $KGE_Q$, $VE_Q$, $NSE_{FDC}$) was lower overall for S2 and S4 than for S1 for both calibration and evaluation periods (Fig. 7), the median NSE runoff ratio ($NSE_{RUNOFF}$) was significantly higher for S4 than for S1 for the evaluation period (Fig. 7b). In contrast, the performance was significantly higher for S3 than for S1 based on median $NSE_{logQ}$ and $VE_Q$ metrics for the calibration period and on median $NSE_Q$, $NSE_{logQ}$, $VE_Q$ and $NSE_{RUNOFF}$ metrics for the evaluation period. These results suggest that simultaneously evaluating model predictions of streamflow and $NO_3^-$ concentration improves the model's ability to reproduce streamflow, especially low flows, due to the improvement in $NSE_{logQ}$. Compared to S1, the model's hydrological performance decreased the most for S2 and the least for S3. The hydrological metrics for S2 also had wider ranges than those for the other scenarios.

Including DOC concentration with streamflow in the calibration showed lower performance for S4 than for S2, while that using $NO_3^-$ concentration showed lower performance for S4 than for S3 (Fig. 7). These results, consistent for both calibration and evaluation periods, supported the observations (Figs. 5 and 6), which suggests that calibrating the model with each solute individually with streamflow better reproduced solute concentrations than calibrating the model with all solutes and streamflow simultaneously.

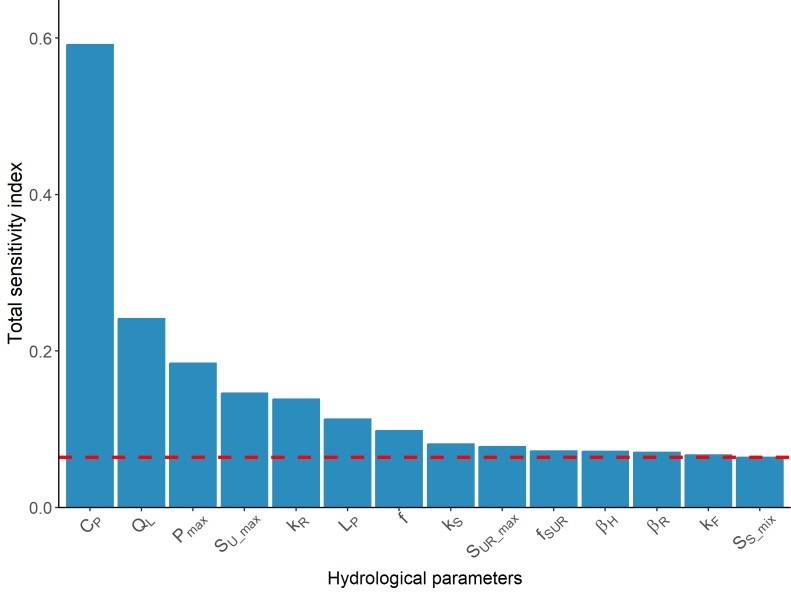

**Figure 3**. Total sensitivity indices estimated using the Fourier Amplitude Sensitivity Test of the influence of hydrological parameters on predicted streamflow. The red dashed line represents the minimum total sensitivity index.

### 3.3. Effects on the distribution of hydrological parameters

Overall, the posterior distribution of hydrological parameters differed among the four calibration scenarios (Fig. 8), except for $f_{SUR}$ and $k_R$, which were less sensitive to the calibration method (i.e. similar optimal values and distributions), indicating that they had been identified well (Fig. 8i, n). For some parameters, the distributions differed only for one scenario, such as $S_{U\_max}$ for S3 (Fig. 8a) and $P_{max}$ for S3 (i.e. smaller values and a narrower range of uncertainties compared to other scenarios, considering both the interquartile range and the total whisker range) (Fig. 8d). The latter suggests that calibration using $NO_3^-$ concentration strongly influenced soil parameters, decreasing percolation of water from $S_U$ to $S_S$. Similarly, the distribution of $S_{UR\_max}$ for S2 differed from other scenarios and had a narrower range of uncertainties, considering both the interquartile range and the total whisker range. This suggests that calibration using DOC concentration improved identification of $S_{UR\_max}$ (Fig. 8l) and that reservoir $S_{UR}$ needs a lower capacity to reproduce both streamflow and DOC concentrations. In addition, for S4, distributions of the most influential hydrological parameters (i.e. $C_P$ and $Q_L$) (Fig. 8b and 8j), as well as of groundwater parameters $k_S$ and $S_{S\_mix}$, differed from those of the other scenarios. Comparing distributions of the groundwater mixing volume in the slow reservoir ($S_{S\_mix}$) for S2 and S3 showed that its size could be decreased by a factor of ca. 3 when calibrating using $NO_3^-$ concentrations instead of DOC concentrations (Fig. 8h).

Overall, all parameters except for $k_F$ and $k_S$ had lower uncertainty when the model was calibrated using solute concentrations, whether simultaneously or separately (Fig. 8). More specifically, the uncertainty in $\beta_H$, $f_{SUR}$, $S_{S\_mix}$ and $k_R$ decreased for S2, S3 and S4. The uncertainty in $C_P$, $\beta_R$ and $S_{UR\_max}$ decreased for S2 and S3, while that in $P_{max}$ and Lp decreased for S3 and S4. The uncertainty in $S_{U\_max}$ decreased only for S2, while that in f decreased only for S3. For deep-infiltration losses ($Q_L$), only calibration using DOC and $NO_3^-$ concentrations simultaneously (S4) decreased its uncertainty compared to those for other scenarios (Fig. 8j).

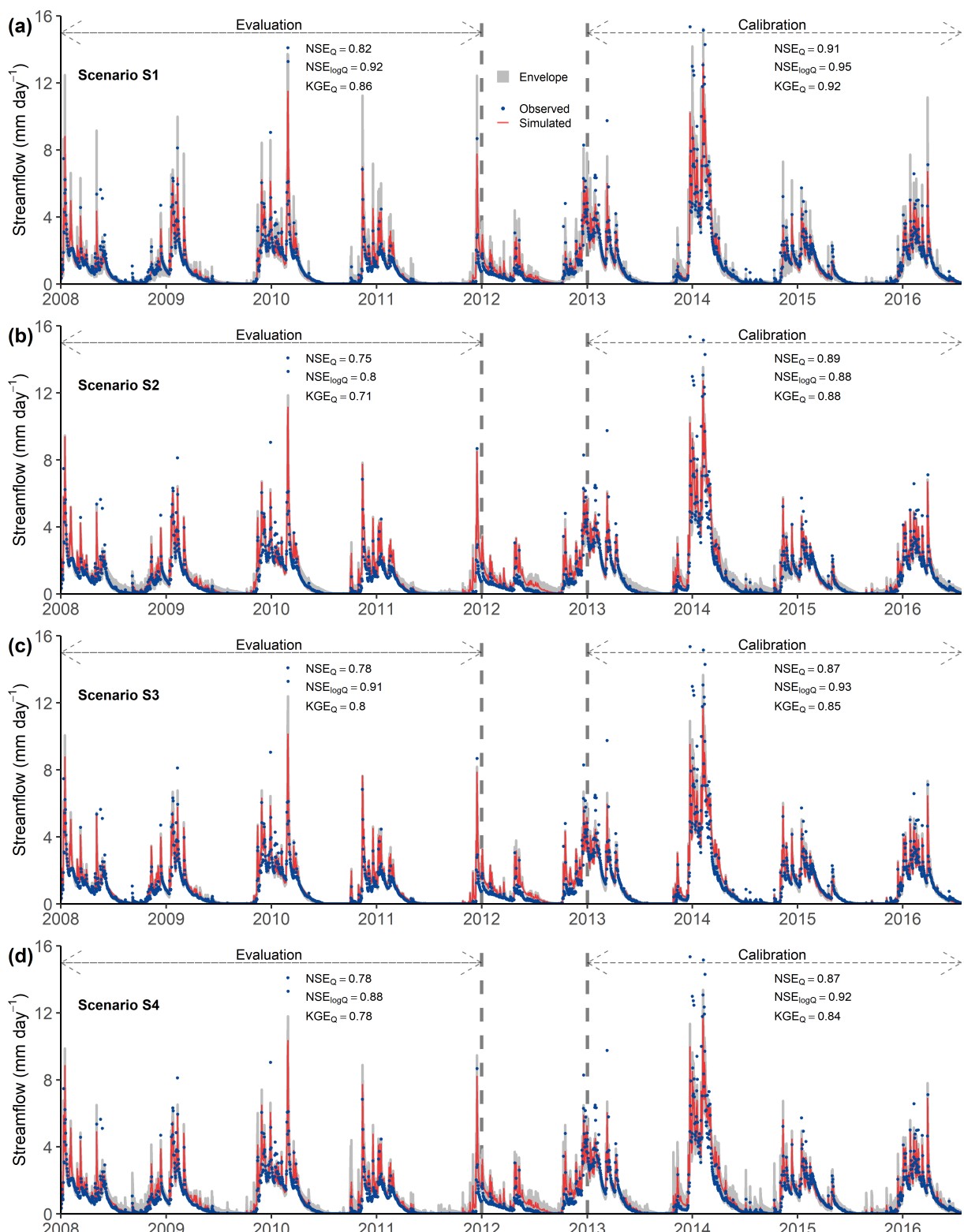

545     **Figure 4**. Observed and simulated flows for the calibration and evaluation periods according to the four scenarios: a) S1 (Hydro only), b) S2 (Hydro + dissolved organic carbon (DOC)), c) S3 (Hydro + nitrate ($NO_3^-$)) and d) S4 (Hydro + DOC + $NO_3^-$). The simulated data for each scenario correspond to the best-compromise simulated discharge of the set of optimal solutions. "Envelope" refers to the simulated discharge envelope using all parameter sets on the Pareto front. See Table 3 and Appendix C for definitions of model-performance metrics.

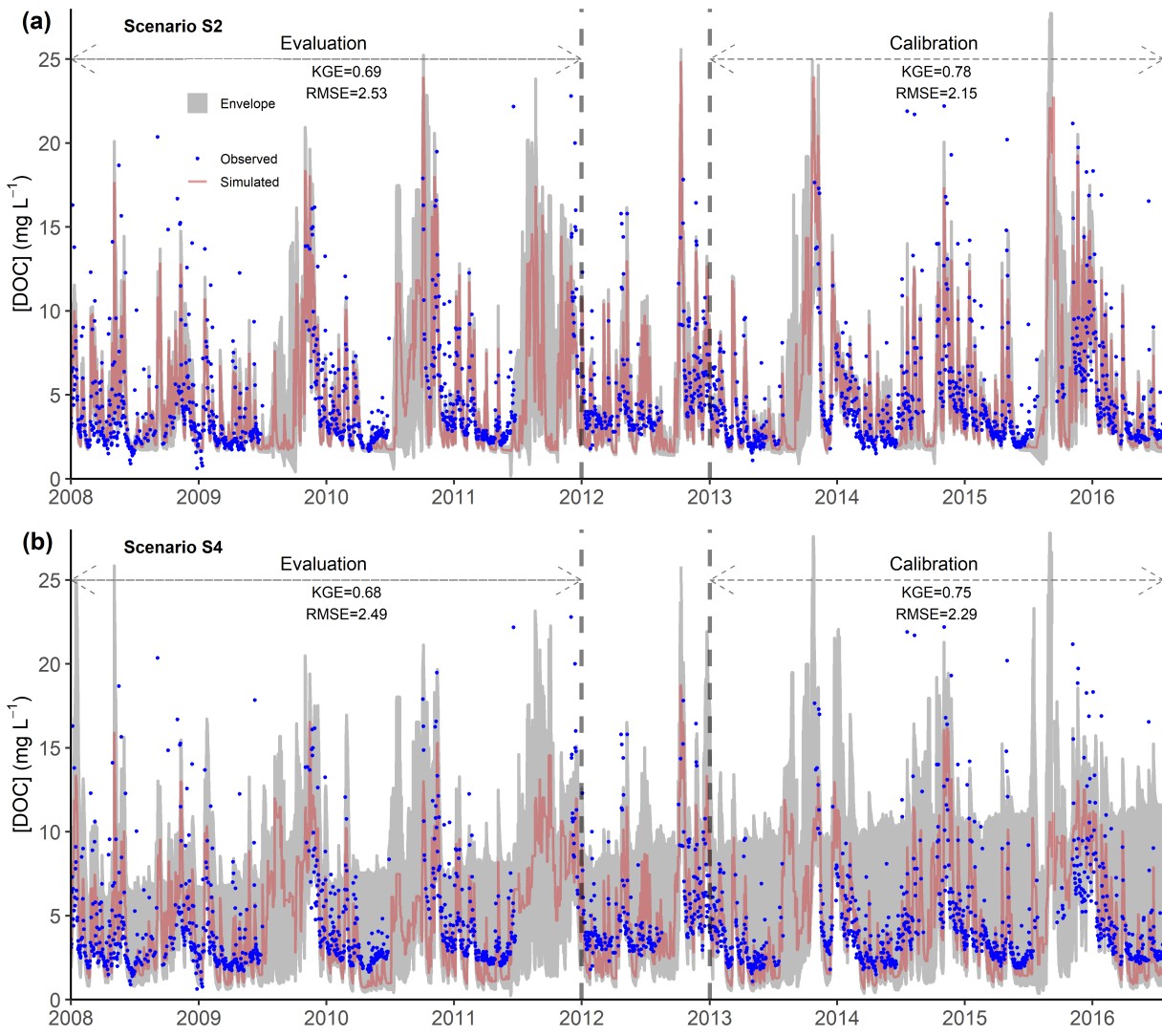

**Figure 5**. Observed and simulated dissolved organic carbon (DOC) concentrations for the calibration and evaluation periods according to two scenarios: a) S2 (Hydro + DOC) and b) S4 (Hydro + DOC + $NO_3^-$). The mean (± standard deviation) observed DOC concentration was $4.8 \pm 3.5$ and $4.5 \pm 3.1$ mg DOC $L^{-1}$ for the calibration and evaluation period, respectively. The simulated data for each scenario correspond to the best-compromise simulated DOC concentration of the set of optimal solutions. "Envelope" refers to the simulated DOC concentration envelope using all parameter sets on the Pareto front. KGE: Kling–Gupta efficiency, RMSE: Root-mean-square error. See Table 3 and Appendix C for definitions of model-performance metrics.

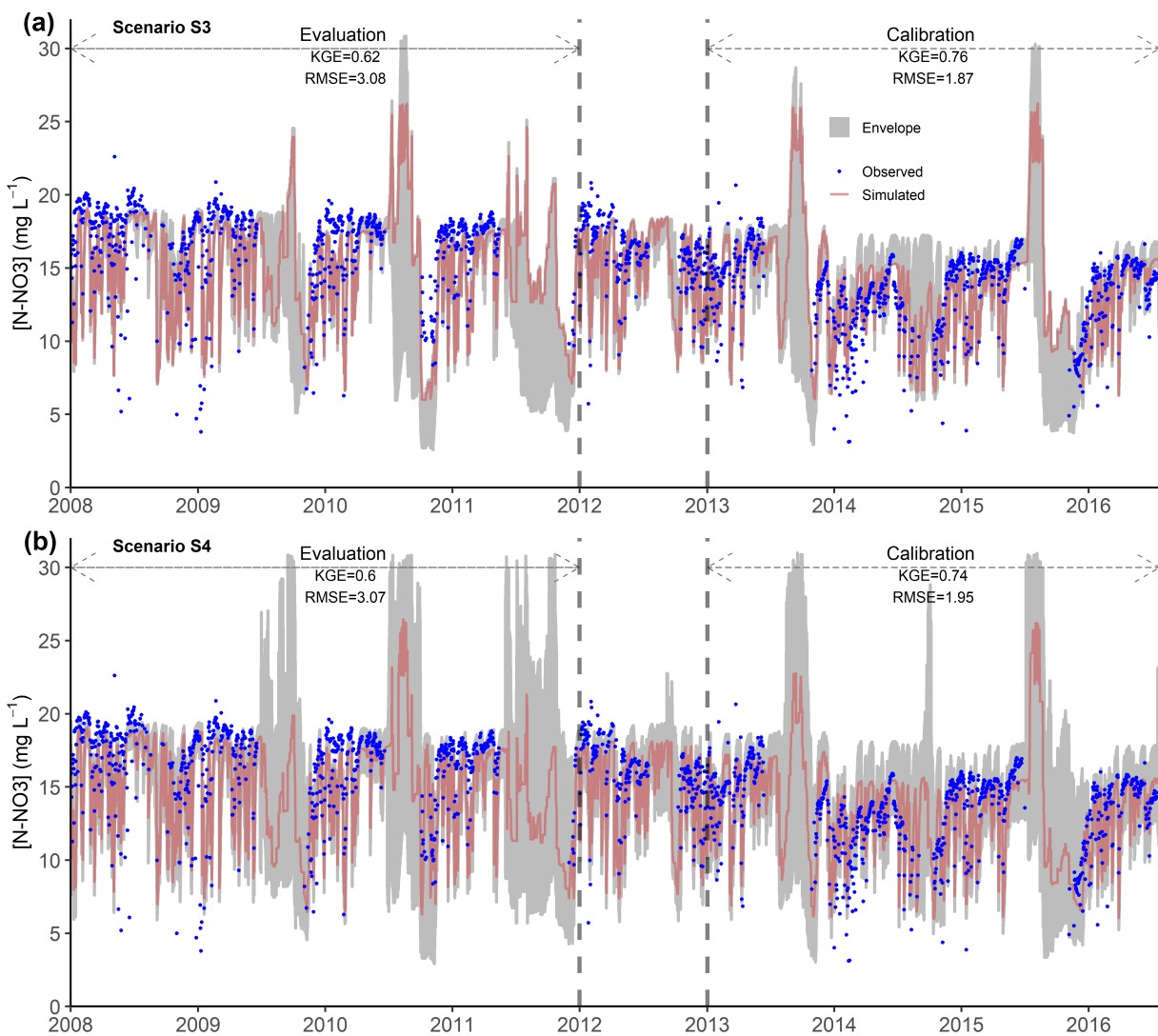

**Figure 6**. Observed and simulated nitrate ($NO_3^-$) concentrations for the calibration and evaluation periods according to two scenarios: a) S3 (Hydro + $NO_3^-$) and b) S4 (Hydro + DOC + $NO_3^-$). The mean (± standard deviation) observed $NO_3^-$ concentration was 13.4 ± 2.7 and 16.6 ± 2.8 mg N-NO$_3$ L$^{-1}$ for the calibration and evaluation period, respectively. The simulated data for each scenario correspond to the best-compromise simulated $NO_3^-$ concentration of the set of optimal solutions. "Envelope" refers to the simulated $NO_3^-$ concentration envelope using all parameter sets on the Pareto front. KGE: Kling–Gupta efficiency, RMSE: Root-mean-square error. See Table 3 and Appendix C for definitions of model-performance metrics.

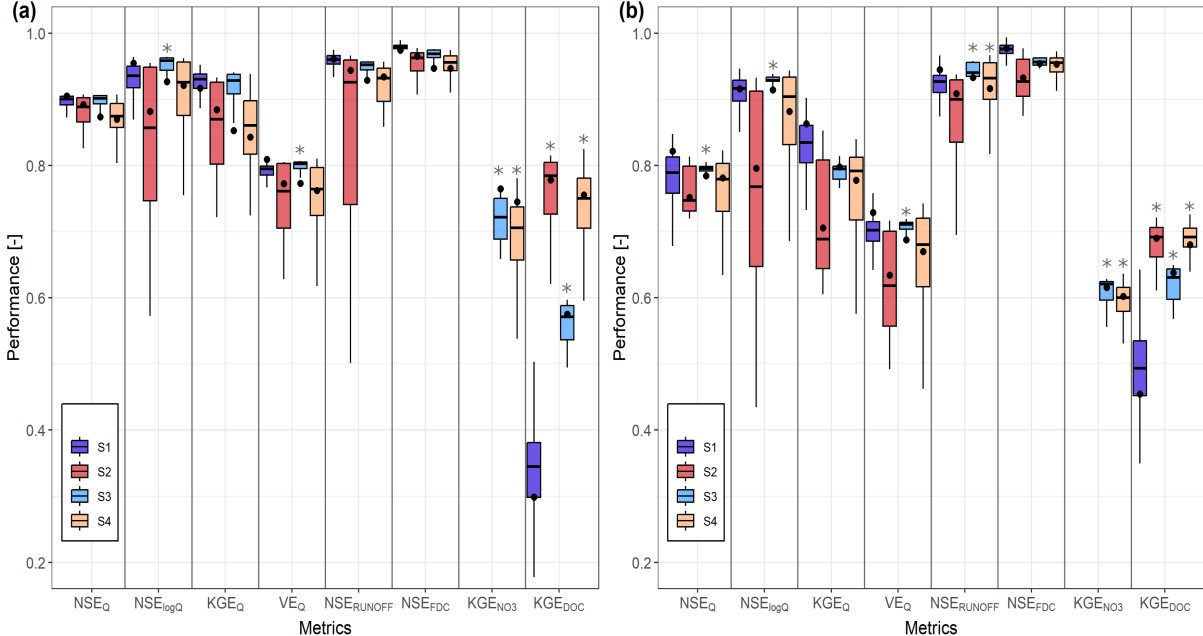

**Figure 7**. Boxplots of performance metrics for predictions of hydrological and solute concentration according to four scenarios: S1 (Hydro Only), S2 (Hydro + DOC), S3 (Hydro + $NO_3^-$) and S4 (Hydro + DOC + $NO_3^-$) for the a) calibration period and b) evaluation period. Whiskers represent 1.5 times the interquartile range. Black circles indicate the best-compromise solution of the Pareto front. The boxplots of $KGE_{NO3}$ for scenarios S1 and S2 are not shown, as their values were negative (median = -1) because the model was not calibrated to represent $NO_3^-$ concentrations in these scenarios. An asterisk above a boxplot indicates values significantly ($p < 0.05$) larger than those for scenario S1 (one-sided Mann-Whitney test). See Table 3 and Appendix C for definitions of model-performance metrics.

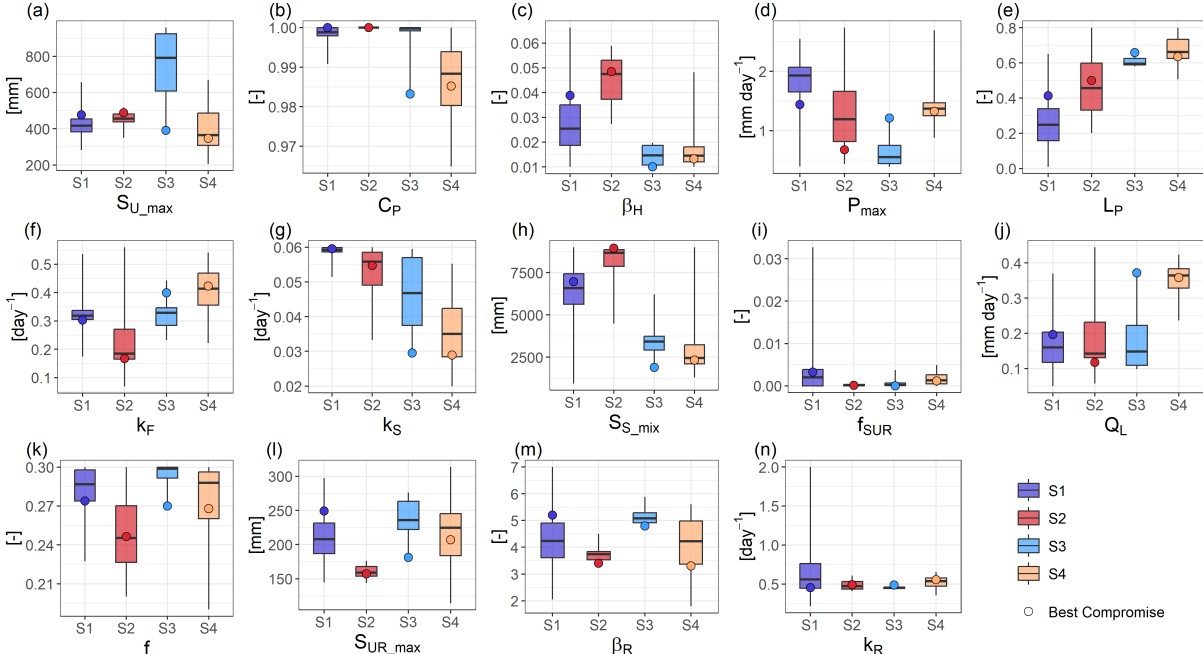

**Figure 8**. Boxplots of hydrological parameters values for the four scenarios: S1 (Hydro Only), S2 (Hydro + DOC), S3 (Hydro + $NO_3^-$) and S4 (Hydro + DOC + $NO_3^-$). Whiskers represent 1.5 times the interquartile range. The circle on each boxplot indicates the parameter's value in the best-compromise set on the Pareto front for each scenario.

### 3.4. Internal model states and consistency

#### 3.4.1. Groundwater level

Overall, the calibration that included solute concentrations with streamflow (S2, S3 and S4) significantly improved simulation of groundwater level compared to S1 (Fig. 9). S1, performance metrics NSE and KGE were indeed the lowest, and PBIAS and RMSE were the highest. S3 and S4 reproduced groundwater levels (NSE = 0.92 and 0.93, respectively) better than S2, while S3 reproduced best the low groundwater levels in 2009, 2011 and 2013. However, for S3 and S4, the model tended to slightly overestimate the low groundwater levels in 2010 and 2015. Overall, the model reproduced the observed magnitude and seasonality of the groundwater level relatively well (NSE = 0.76-0.93, depending on the scenario). PBIAS values were negative for all scenarios, indicating that the model tended to underestimate groundwater level.

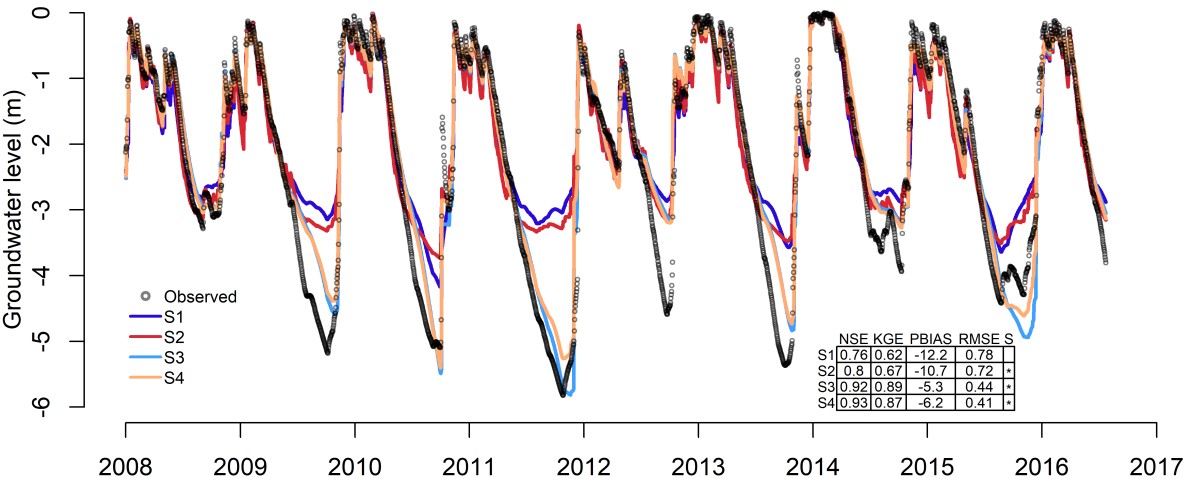

**Figure 9**. Observed and simulated groundwater levels for the four scenarios: S1 (Hydro Only), S2 (Hydro + DOC), S3 (Hydro + $NO_3^-$) and S4 (Hydro + DOC + $NO_3^-$). NSE: Nash–Sutcliffe model efficiency coefficient, KGE: Kling–Gupta efficiency, PBIAS = Percent bias, RMSE: Root-mean-square error, S: Significance level. An asterisk in the significance level column indicates values that differed significantly ($p < 0.05$) from those for scenario S1 (two-sided Mann-Whitney test). See Appendix C for definitions of the performance metrics.

#### 3.4.2. Soil moisture

Overall, calibrating the model with streamflow and solute concentrations simultaneously did not improve simulation of soil moisture dynamics in the riparian zone compared to S1 (Fig. 10a). The calibration that included DOC concentrations with streamflow (S2) had significantly lower performance to reproduce normalized soil moisture at PG2 (NSE = 0.58 and KGE = 0.74) compared to S1. The model reproduced observed soil moisture better when it was calibrated using DOC and $NO_3^-$ simultaneously (S4, with NSE = 0.73 and KGE = 0.78) than when using only one solute (S2 or S3, with NSE = 0.58 and 0.69, respectively, and KGE = 0.74 and 0.75, respectively). The model reproduced major features of the observed dynamics of normalized soil moisture at PG2 (i.e. the riparian zone) (NSE = 0.58-0.79, depending on the scenario). It also reproduced drying rates at the end of the summer and wetting rates well overall. However, the model tended to slightly underestimate soil moisture in summer 2015 and winter 2016. PBIAS values were negative for all scenarios, indicating that the model tended to underestimate normalized soil moisture at PG2.

Only S2 reproduced soil moisture in the upslope zone significantly better than S1 did (NSE = 0.94 and 0.92, respectively) (Fig. 10b). For S3 and S4, the model did not reproduce the wetting rate well at the beginning of 2017, when it overestimated soil moisture. S3 and S4 had significantly lower performance than S1 did. Overall, the model reproduced the observed dynamics of normalized soil moisture at Toullo (i.e. the upslope zone) (NSE = 0.79-0.94, depending on the scenario).

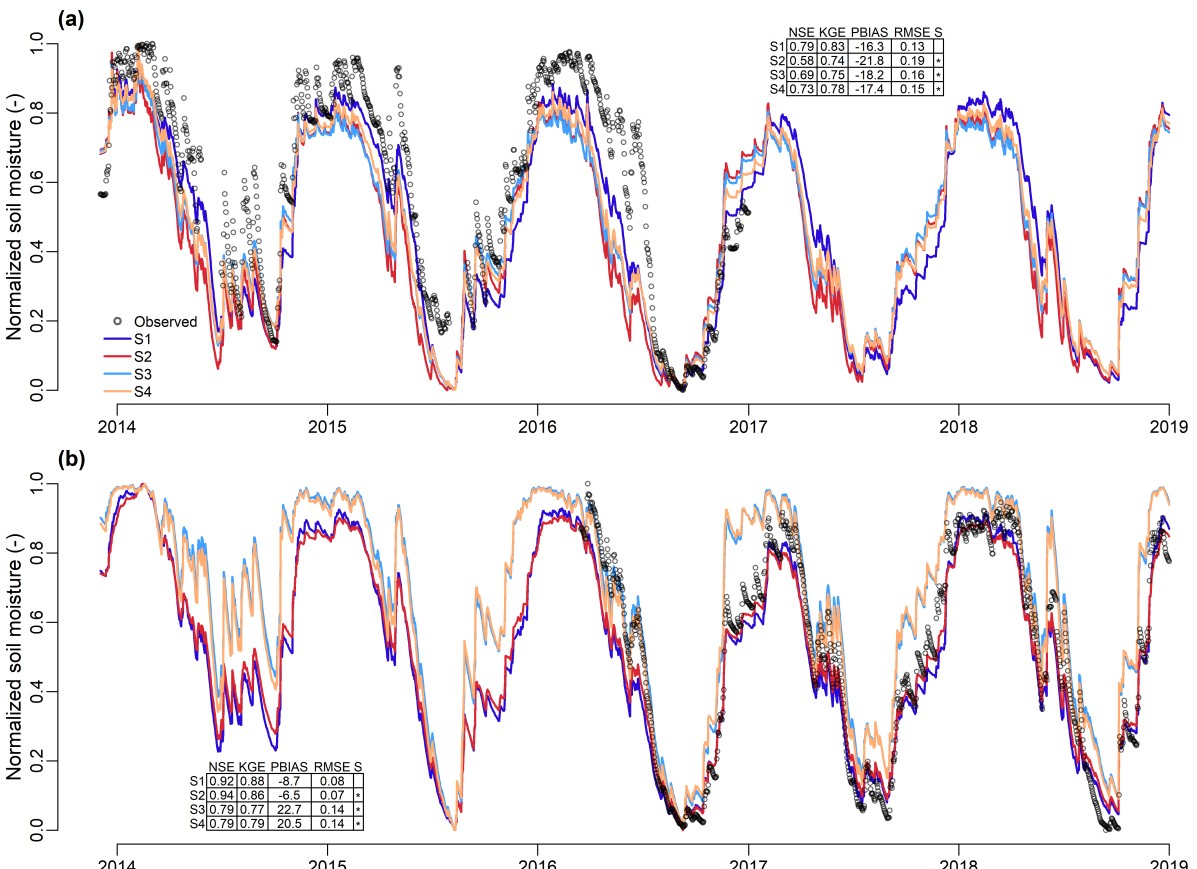

**Figure 10**. a) Normalized observed (point PG2) and simulated soil moisture in the $S_{UR}$ reservoir and b) Normalized observed (Toullo point) and simulated soil moisture in the $S_U$ reservoir for four calibration scenarios: S1 (Hydro Only), S2 (Hydro + DOC), S3 (Hydro + $NO_3^-$) and S4 (Hydro + DOC + $NO_3^-$). NSE: Nash–Sutcliffe model efficiency coefficient, KGE: Kling–Gupta efficiency, PBIAS = Percent bias, RMSE: Root-mean-square error, S: Significance level. An asterisk in the significance level column indicates values that differed significantly ($p <$ 0.05) from those for scenario S1 (two-sided Mann-Whitney test). See Appendix C for definitions of the performance metrics.

## 3.5. Water balances

Calibrating the model with DOC and $NO_3^-$ concentrations along with streamflow data influenced water-balance components and changed the storage in reservoirs $S_U$, $S_S$ and $S_{UR}$. Median simulated total evaporative flow ($E_U$ and $E_{UR}$) was highest for S1 (470 mm yr$^{-1}$) and lowest for S4 (372 mm yr$^{-1}$) (Fig. 11a). Median deep-infiltration losses ($Q_L$) were highest for S4 (128 mm yr$^{-1}$) and lowest for S3 (54 mm yr$^{-1}$). The median contribution of $S_R$ to discharge ($Q_R$) was slightly but significantly higher for S3 and S4 (108 and 109 mm yr$^{-1}$, respectively) than for S1 (100 mm yr$^{-1}$). The median contribution of $S_S$ to discharge ($Q_S$) was significantly higher for S2 (293 mm yr$^{-1}$) than for S1 (242 mm yr$^{-1}$). $S_S$ and $S_{UR}$ stored water during the simulation, while $S_U$ lost water. $S_S$ tended store

significantly more water for S4 (2.7 mm yr$^{-1}$) than it did for S1 (1.2 mm yr$^{-1}$) (Fig. 11b). $S_U$ lost significantly more water for S3 (-21 mm yr$^{-1}$) than for S1 (-12 mm yr$^{-1}$) and lost the least for S4 (-10.6 mm yr$^{-1}$).

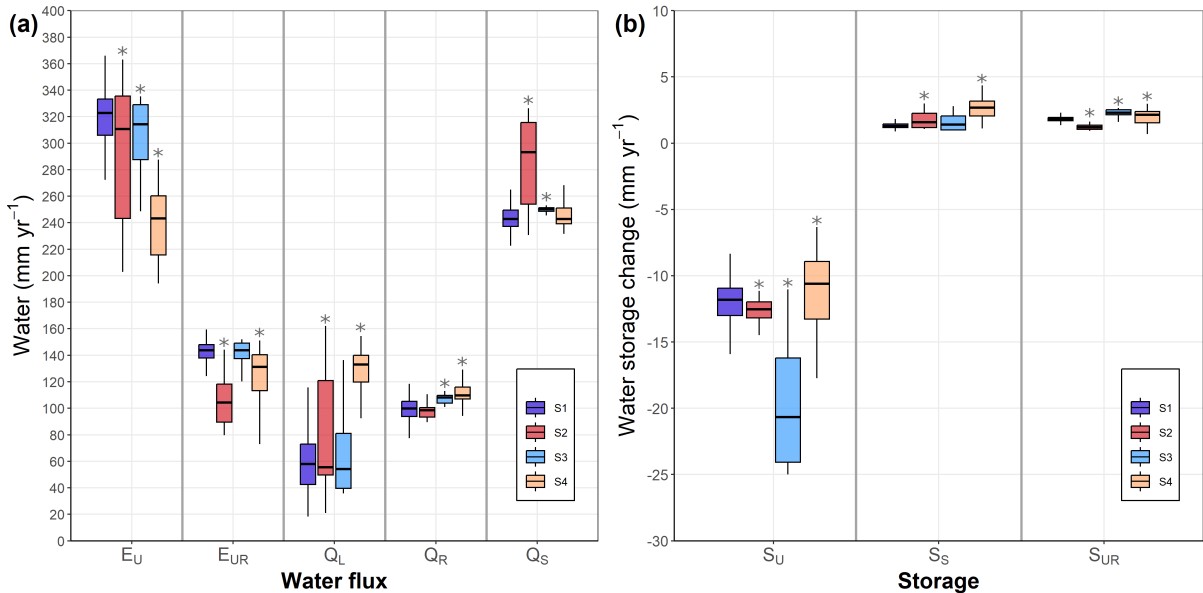

**Figure 11**. a) Boxplots of simulated annual water budgets for all Pareto fronts of each scenario (S1-S4) during the calibration and evaluation periods combined (1 Aug 2008-1 Sep 2016). Boxplots of changes in simulated storage of the main reservoirs of the model for all Pareto fronts of each scenario during the period. Whiskers represent 1.5 times the interquartile range. An asterisk above a boxplot indicates values that differed significantly ($p < 0.05$) from those for scenario S1 (two-sided Mann-Whitney test).

## 4. Discussion

### 4.1. Effect on streamflow, groundwater and soil moisture

The parsimonious solute-transport model reasonably reproduced simultaneously the dynamics of discharge, DOC and $NO_3^-$ concentrations in the stream of the Kervidy-Naizin catchment for all scenarios. Model predictions based on independent data indicated that the model generally reproduced the dynamics of groundwater level and soil moisture in upslope and riparian zones for all scenarios. Including solute (DOC and $NO_3^-$) data along with streamflow data in a multi-objective calibration strategy improved the representation of groundwater storage and soil moisture in the upslope zone (Figs. 9 and 10b). The improvement in the representation of groundwater level was significant and relatively large for scenarios S2, S3 and S4 compared to S1 (Fig. 9). In contrast, the improvement in the representation of soil moisture in the upslope zone was significant but relatively small only for scenario S2 compared to S1 (Fig. 10b). Thus, only scenario S2 improved the representation of both groundwater and soil moisture in the upslope zone.

Studies have shown that using additional information to constrain hydrological models usually improves spatial and/or temporal patterns of internal state variables and flows but does not necessarily improve the accuracy of predicted runoff (López López et al., 2017; Tong et al., 2021). Woodward et al. (2013b) developed a catchment simulation model that predicted streamflow and water chemistry by connecting a model of soil water balance to two groundwater reservoirs. They found that calibrating the model using daily streamflow and monthly $NO_3^-$ data simultaneously from a small lowland milk-production-oriented catchment improved hydrological understanding and estimated catchment $NO_3^-$ flows relatively well. In particular, they were able to infer daily contributions of

near-surface water, fast shallow groundwater, and slower, deeper groundwater to water and $NO_3^-$ discharge. However, including $NO_3^-$ data in the calibration overpredicted low flows compared to calibration using streamflow data alone. Yen et al. (2014) used regional estimates of annual denitrification mass and the percentage of $NO_3^-$ load at the catchment outlet that had come from groundwater as soft data to constrain water-flow partitioning, which yielded realistic internal catchment behaviour but decreased the accuracy of predicted streamflow. In this study, when considering only the best-compromise model for each scenario, the use of solute data improved the representation of groundwater storage (S2, S3 and S4, Fig. 9) and soil moisture in the upslope zone (S2, Fig. 10b), but slightly decreased the accuracy of predicted streamflow in both calibration and evaluation periods (Fig. 4). In contrast, considering all hydrological signatures for discharge obtained from the envelope, S3 improved the model's ability to reproduce streamflow characteristics, especially low flows (Fig. 7) and groundwater level (Fig. 9).

We included solutes (DOC and $NO_3^-$) that have opposite dynamics and whose conceptual models had been successfully tested in the literature (Birkel et al., 2014; Fovet et al., 2015b), with the aim of adding useful constraints to the hydrological modelling. However, none of the scenarios that included DOC and/or $NO_3^-$ improved both the model's representation of streamflow dynamics and internal consistency in representing groundwater level and soil moisture in the riparian and upslope zones. Given the limits of this study, it remains uncertain whether including solutes with streamflow in calibration improved only the representation of hydrological states and flows of specific reservoirs or also improved the model's overall internal consistency. The first limit came from comparing point-scale in-situ observations to simulated soil moisture and groundwater levels that represented catchment-scale storage, as these observations may not have represented the actual dynamics of groundwater and soil storage. Furthermore, although the dynamics of DOC and $NO_3^-$ concentrations in the stream were represented well, the conceptualization of biogeochemical processes and transport of these solutes may remain too simple to represent internal state variables and flows of solutes. The model represents the hydrological and biogeochemical processes that are assumed to dominate, and these assumptions are limited by incomplete knowledge. In addition, the representation of reactive solutes increased the number of parameters and the complexity of the model. Consequently, it would be interesting to compare this approach to the use of non-reactive solutes in calibration, such as natural tracers that are assumed to be conservative, including chloride ($Cl^-$) and stable isotopes of water ($^{18}O$ and $^2H$) (Kirchner et al., 2010), to assess whether the model can reproduce the dynamics of both soil moisture and groundwater better.

The factors that improve internal hydrological consistency when solute data are included are not well understood. Streamflow aggregates information from many catchment-scale processes, but this information is too ambiguous to determine the exact catchment configuration (Kuppel et al., 2018b) or flow pathways that produced the observed signal (Woodward et al., 2017). This is because streamflow aggregates downstream along a convergent network towards a single outlet, but the divergent nature of an upstream network makes it impossible to uniquely backtrack the locations where the flow was generated (Kirchner et al., 2001). Thus, streamflow can be simulated well with many alternative model parameterizations, whether or not they are physically consistent (Kirchner, 2006). Results of the present study thus suggest that if streamflow alone is used for calibration, the model predicts discharge correctly for the wrong reason, as internal consistency, especially the representation of groundwater level, is not guaranteed. The model thus simulates water pathways and storage dynamics that do not represent those in the actual catchment. Consequently, it appears that the hydrological behaviour of the catchment required to reproduce the observed DOC and $NO_3^-$ concentrations in the stream is different from that required to reproduce only the observed discharge. This hypothesis is supported by the fact that the calibration scenarios influenced the main

components of the water balance differently. For example, S3 yielded better representation of the groundwater reservoir, with good reproduction of the groundwater level (Fig. 9), but lower evapotranspiration and higher water loss from the $S_U$ reservoir than S1 (Fig. 12). In comparison, S2 yielded better representation of upslope soil water storage (Fig. 10b) and a higher contribution of $S_S$ to discharge than S1 (Fig. 12). The large amount of information in the solute time series thus constrained internal reservoirs and water flows more than a streamflow-only approach, which increased internal consistency of the hydrological model. This occurs because a hydrological model needs to represent only an input-output response, whereas when biogeochemistry is included, a model needs to represent both residence-time distributions and biogeochemical processing to reproduce the observed stream concentrations (Medici et al., 2012) and the decrease in solute-input signals. The use of solute time series, which mitigates the equifinality problem, thus excluded infeasible model configurations that would have also yielded high performance (Dimitrova-Petrova et al., 2020; Kuppel et al., 2018b; Yen et al., 2014).

An additional step is needed to understand the benefits of including solute data for internal hydrological consistency by analysing effects of including DOC and $NO_3^-$ concentration data on the storage dynamics (state and flows) of model components. For example, the simulations showed that including $NO_3^-$ data decreased $k_S$ and $S_{S\_mix}$ (Fig. 8g and 8h), suggesting that simulations of $NO_3^-$ dynamics were optimized at a lower groundwater mixing volume and lower flow rate in $S_S$. However, it is important to go further to understand why including $NO_3^-$ concentration data improved simulation of groundwater level (Fig. 9) and low flow (Fig. 7). In this landscape, most of the $NO_3^-$ leached from the unsaturated reservoir accumulates in the shallow groundwater (Aubert et al., 2013; Strohmenger et al., 2020). The groundwater, with a legacy mass storage of $NO_3^-$ (Basu et al., 2010; Molenat et al., 2008), thus contributes water to the stream that sustains the base flow and export of $NO_3^-$ (Aubert et al., 2013; Molenat et al., 2008). Given these characteristics, good reproduction of $NO_3^-$ concentrations and flows in the stream, supplied mainly by groundwater, can be assumed to constrain the model sufficiently to yield good reproduction of water flows from the groundwater to the stream and thus good representation of groundwater level.

### 4.2. Effects on parameter uncertainties

Using a parsimonious hydro-chemical model without explicit biogeochemical processes, Strohmenger et al. (2021) found that overall parameter uncertainties were higher when calibrating using solute data (DOC, $NO_3^-$) along with streamflow data than when calibrating using streamflow data alone. They assumed that DOC and $NO_3^-$ sources behave as infinite pools with a fixed concentration in each reservoir contributing to the stream. The modelling approach in the present study was relatively similar, but explicitly represented biochemical processes related to DOC and $NO_3^-$. This approach resulted in decreased parameter uncertainty when solute concentrations were included in calibration, except for storage coefficients of the fast ($k_F$) and slow reservoirs ($k_S$) (Fig. 8). Comparing the results of these two studies suggests that the infinite-solute-pool assumption is sufficient to reproduce annual and storm-event dynamics of discharge and DOC and $NO_3^-$ concentrations in the stream, but is insufficient in calibration to constrain the model to adequately reproduce water storage dynamics and flow paths and to reduce uncertainties in hydrological parameters. In the infinite-solute-pool assumption, hydrological parameters are indeed less sensitive to solute concentrations than they are in models that explicitly represent biogeochemical processes and dynamic solute concentrations in reservoirs. Notably, the results of this study highlight that S4, which considered all constraints (i.e. streamflow and DOC and $NO_3^-$ concentrations), greatly influenced the distributions of the most influential hydrological parameters, specifically $Q_L$ and $C_P$, whose values were among the highest or lowest, respectively (Fig. 8b and 8j), and reproduced groundwater levels the best (Fig. 9). This highlights the importance of parameters $Q_L$ and $C_P$, which determine inter-catchment groundwater flows and the

recharge from $S_U$ to $S_S$ and $S_{UR}$ to $S_R$, respectively, in ensuring that the model reproduced the observed groundwater dynamics. Based on these results, for the model to best reproduce the dynamics of streamflow, concentrations (DOC, $NO_3^-$) and groundwater, recharge should be decreased and inter-catchment groundwater flow should be increased to ca. 0.35 mm day$^{-1}$ (best-compromise parameter value for S4, Fig. 8j). This value is consistent with those found in modelling studies of a similar physiographic headwater catchment in Brittany (Fovet et al., 2015a;

Hrachowitz et al., 2014).

The model conceptualizes biogeochemical processes for DOC and $NO_3^-$ in a relatively simple way, but has reduced the uncertainties of the parameters. An additional step in future studies will be to analyse whether more complex representation of biogeochemical processes in the model can further reduce uncertainties in hydrological parameters. Results of the present study are consistent with those of other studies, in which inclusion of additional

variables in multiple-objective calibration generally reduced parameter uncertainty (Tong et al., 2021). For example, Yen et al. (2014) found that including data related to water quality yielded lower parameter uncertainties than calibration using streamflow alone, especially for hydrological parameters that strongly influence denitrification. Silvestro et al. (2015) demonstrated that the equifinality of soil parameters was reduced by including satellite-derived soil moisture when calibrating a process-based, spatially distributed hydrological

model. Similarly, Rajib et al. (2016) found that including satellite-derived soil moisture, especially that in the rooting zone, reduced parameter uncertainties, particularly for parameters related to subsurface hydrological processes.

**4.3. Comparability of point-scale *in-situ* measurements to catchment-scale storage**

A remaining issue is the limited comparability of point-scale *in-situ* measurements and simulated soil moisture

and groundwater level to catchment-scale storage. In-situ volumetric soil moisture was calculated as the mean of several TDR probes, which reduces uncertainty at the point scale, but upscaling these point measurements to a reservoir that represents a hillslope or riparian zone is associated with uncertainties. Consequently, we considered normalized soil moisture as a proxy for dynamics of unsaturated storage in hillslope and riparian zones. Similarly, we used the daily mean normalized water level at point PG5 as a proxy for groundwater storage dynamics. An

additional step in future studies will be to determine how point measurements can be upscaled to areal mean point scale soil moisture and groundwater measurements compatible with catchment-scale storage. A complementary approach is to include other promising methods, such as remote sensing, to estimate the spatial distribution of storage in catchments, especially of soil moisture (Duethmann et al., 2022; Tong et al., 2021). The high spatial resolution, worldwide spatial coverage and increasing availability of remotely sensed data may provide ample

opportunities to further constrain hydrological models and their parameters (Bouaziz et al., 2021; Duethmann et al., 2022; Gomis-Cebolla et al., 2022; Nijzink et al., 2018; Tong et al., 2021). Recent soil moisture data from satellite-derived soil-moisture products (e.g. SMAPL3E, SCATSAR, ASCAT DIREX SWI) with high spatial and temporal resolutions (e.g. 0.5-9.0 km and 1-3 days, respectively) (Duethmann et al., 2022) would help constrain the model of the Kervidy-Naizin catchment. Other promising methods include cosmic-ray neutron-sensor probes

to estimate dynamics of near-surface soil water storage (Dimitrova-Petrova et al., 2020) and geodesy and geophysical methods (Fovet et al., 2015a). Additional data can be used to assess the internal representation of evapotranspiration, which has a wide spatial and temporal distribution at the catchment scale, to provide more confidence in simulation of the partitioning of water between soil storage and groundwater recharge (Moazenzadeh and Izady, 2022). For example, using spatially and temporally gridded remotely sensed evapotranspiration data to

calibrate the Soil and Water Assessment Tool (SWAT) hydrological model decreased the equifinality of the

calibrated parameters compared to calibration using only streamflow data (Shah et al., 2021). These results demonstrate the benefit of using increasingly available open-access remotely sensed evapotranspiration data to improve calibration of hydrological models. These methods provide a spatially aggregated overview of catchment water content and go beyond traditional methods of direct storage observations at the point scale that are limited to a single reservoir (Dimitrova-Petrova et al., 2020).

## 4.4. Implications

This study's results indicate that solute data are important for improving the internal consistency of hydrological models, which can help guide collection of field data and modelling (Stadnyk and Holmes, 2023). When collecting field data for model calibration, it may be important to collect solute data along with streamflow data. These data can then be used in a hydrological model to which simple representations of biogeochemical processes are added to improve the representation of internal behaviour of the catchment by calibrating streamflow and solutes simultaneously. The type of solute measured is also important, as calibration using $NO_3^-$ improved the internal consistency of the groundwater reservoir, while that using DOC improved the internal consistency of soil water storage in the upslope zone. With the increasing availability of solute data from catchment monitoring, this approach provides an objective way to improve representation of complex hydrological systems when information about their internal functioning is insufficient. A catchment model that represents observed behaviour of the system more accurately can then be used with confidence when assessing scenarios, such as those of nutrient remediation or climate change. If the internal behaviour of the hydrological system is not represented correctly, predicting streamflow acceptably is pointless and perhaps counter-productive, leading to erroneous conclusions and potential mismanagement of catchment resources. For example, Yen et al. (2014) showed that a lack of constraints to realistically represent the internal functioning of a catchment can lead to misleading assessments of pollution-control scenarios, even when typical streamflow performance criteria are satisfied.

The ability to apply this modelling approach to other catchments with different physiography will depend on the model's ability to represent dominant sources and pathways of DOC and $NO_3^-$ concentrations that differ from those of Kervidy-Naizin. To address this question, we analysed the response of streamwater chemistry to changes in discharge observed in this catchment and how the model represents it. Changes in solute concentrations as a function of discharge (i.e. concentration-discharge (CQ) relationships) (Appendix B) provide insight into how catchments store and release water and solutes, and can therefore be considered a "fingerprint" of catchment transport, mixing and reaction processes (Godsey et al., 2009; Knapp et al., 2020). Long-term seasonal CQ slopes for $NO_3^-$ in Kervidy-Naizin generally indicated a chemostatic $NO_3^-$ export regime (Fig. B1a). Indeed, this pattern often emerges in catchments with a spatially uniform distribution of abundant solute sources, such as $NO_3^-$ in agricultural areas, which leads to a relatively constant release of solutes to the stream network (Bieroza et al., 2018). In contrast, in the winter of a few years (2010, 2012 and 2014), the CQ slope indicates instead a slightly more chemodynamic export regime with a dilution pattern. Long-term seasonal CQ slopes for DOC indicate a chemodynamic export regime with an accretion pattern that changes to a chemostatic export regime in autumn (Fig. B1b). The model reproduced the differing export regime of each solute from 2008-2016 relatively well (Fig. B1). Model performance was slightly lower for DOC (RMSE = 0.13-0.27) than for $NO_3^-$ (RMSE = 0.08-0.21). For a few years, the model did not represent the export regime accurately. The export regime for $NO_3^-$ observed in winter 2008 and 2009 was chemostatic, but the model simulated a chemodynamic export regime with a dilution pattern. The export regime for DOC observed in autumn 2011 and 2014 and summer 2012 and 2014 was an chemostatic export regime, but the model simulated a more chemodynamic export regime with an accretion

pattern. As the model simulated hydrological dynamics relatively well during these periods (Fig. 4), it was likely overpredicting DOC. Analysis of the CQ relationships observed and simulated in Kervidy-Naizin highlighted two important points: (i) each solute in this catchment did not have a single pattern but instead seasonal and interannual differences in export regimes and (ii) the parsimonious solute-transport model was able to reproduce different export regimes. Thus, this modelling approach may be applicable, in particular due to its flexible structure, to headwater catchments whose characteristics and export regimes differ from those of Kervidy-Naizin. Applying the model to catchments whose streams can be intermittent would first require solving the methodological issue of high $NO_3^-$ concentrations in summer, when no observed data are available, to prevent overpredicting concentrations and risk overestimating $NO_3^-$ flows in summer. The model can also be adapted to represent catchments whose hydrological and biochemical patterns differ from those of Kervidy-Naizin, where most DOC accumulates in the soils of the riparian zone and $NO_3^-$ accumulates in the groundwater. For example, the reservoirs in which DOC is produced or lost can be modified easily. In addition, more complex models of biogeochemical processes can also be considered. While we represented heterotrophic denitrification as a constant, dynamic equations (Heinen, 2006) could easily be incorporated to represent the seasonality of this process.

**5. Conclusion**

The model reasonably reproduced the dynamics of discharge and solute (DOC and $NO_3^-$) concentrations in the stream of the headwater catchment simultaneously for all scenarios. Model predictions based on independent data indicated that the model generally reproduced the dynamics of groundwater level and soil moisture in upslope and riparian zones for all scenarios. Given the performance of the best-compromise model for each scenario, the results of this study tend to reject the first hypothesis, as using daily stream DOC and $NO_3^-$ concentrations along with streamflow data to calibrate the model did not improve the model's performance for simulated streamflow for the calibration or evaluation period compared to calibration with streamflow alone. In contrast, considering all hydrological signatures for discharge obtained from the envelope, the scenario that included $NO_3^-$ along with streamflow improved the model's ability to reproduce streamflow, especially low flows. For the second hypothesis, including solute data along with streamflow data in a multi-objective calibration strategy significantly improved the representation of groundwater storage and soil moisture in the upslope zone. The improvement in the representation of groundwater level was significant and relatively large for all scenarios when using one or both solutes along with streamflow for calibration compared to using only streamflow. In contrast, the improvement in the representation of soil moisture in the upslope zone was significant but relatively small only when using DOC concentration along with streamflow for calibration compared to using only streamflow. None of the scenarios that included solutes improved both the model's representation of streamflow dynamics and internal consistency in representing groundwater level and soil moisture in the riparian and upslope zones. Based on these results, it remains uncertain whether including solutes with streamflow in calibration improved only the representation of hydrological states and flows of specific reservoirs or also improved the model's overall internal hydrological consistency. For the third hypothesis, explicitly modelling biochemical processes for DOC and $NO_3^-$ reduced the uncertainty in hydrological parameters, except the storage coefficients of the fast and slow reservoirs, compared to an approach in which sources of DOC and $NO_3^-$ were treated as infinite pools with fixed concentrations. The simultaneous inclusion of daily in-stream DOC and $NO_3^-$ concentrations in the calibration strategy influenced the distribution of the most influential hydrological parameters of the model. Differences among the calibration scenarios also influenced the main components of the water balance. Calibrating the model

with streamflow and solute concentrations simultaneously reduced predictions of evapotranspiration. Compared to calibration using streamflow alone, the inclusion of DOC increased the predicted contribution of groundwater to discharge, while the inclusion of $NO_3^-$ increased the predicted loss of water from the rooting-zone reservoir. This modelling study demonstrated that including the large amount of information in solute time series in hydrological models provided an objective way to improve the representation of complex hydrological systems for which information about internal functioning was insufficient.

## Appendix A

**Table A1.** Symbols and definitions of variables in the hydrological model

| Symbol | Definition | Symbol | Definition |
|--------|-----------|--------|-----------|
| $C_{H,R}$ | Hillslope runoff coefficient [–] | $Q_T$ | Total outflow [L T$^{-1}$] |
| $C_P$ | Preferential recharge coefficient [–] | $R_F$ | Recharge of fast reservoir [L T$^{-1}$] |
| $C_{R,R}$ | Riparian runoff coefficient [–] | $R_{FR}$ | Recharge of $S_{UR}$ from $S_F$ [L T$^{-1}$] |
| $E_A$ | Actual evaporation [L T$^{-1}$] | $R_P$ | Preferential recharge of the slow reservoir [L T$^{-1}$] |
| $E_P$ | Potential evaporation [L T$^{-1}$] | $R_R$ | Recharge of the riparian zone reservoir [L T$^{-1}$] |
| $E_U$ | Transpiration from $S_U$ [L T$^{-1}$] | $R_{SR}$ | Recharge of $S_{UR}$ from $S_S$ [L T$^{-1}$] |
| $E_{UR}$ | Transpiration from $S_{UR}$ [L T$^{-1}$] | $R_{SS}$ | Recharge of the slow reservoir [L T$^{-1}$] |
| f | Proportion of the catchment covered by the riparian zone [–] | $R_U$ | Infiltration into the unsaturated reservoir [L T$^{-1}$] |
| $f_{SUR}$ | Proportion of water flow from $S_S$ that passes through $S_{UR}$ [–] | $S_F$ | Storage in the fast reservoir [L] |
| $k_F$ | Storage coefficient of the fast reservoir [T$^{-1}$] | $S_R$ | Storage in the riparian reservoir [L] |
| $k_R$ | Storage coefficient of the riparian zone reservoir [T$^{-1}$] | $S_S$ | Storage in the slow reservoir [L] |
| $k_S$ | Storage coefficient of the slow reservoir [T$^{-1}$] | $S_{S\_mix}$ | Groundwater mixing storage in the slow reservoir [L] |
| $L_P$ | Transpiration threshold [–] | $S_U$ | Unsaturated storage [L] |
| P | Precipitation [L T$^{-1}$] | $S_{U\_max}$ | Storage capacity of the hillslope unsaturated zone [L] |
| $P_{max}$ | Percolation capacity [L T$^{-1}$] | $S_{UR}$ | Unsaturated storage in the riparian zone [L] |
| $Q_L$ | Deep infiltration loss [L T$^{-1}$] | $S_{UR\_max}$ | Storage capacity in the riparian unsaturated zone [L] |
| $Q_R$ | Runoff from the riparian zone reservoir [L T$^{-1}$] | $\beta_H$ | Hillslope coefficient [–] |
| $Q_S$ | Runoff from the slow reservoir [L T$^{-1}$] | $\beta_R$ | Riparian coefficient [–] |

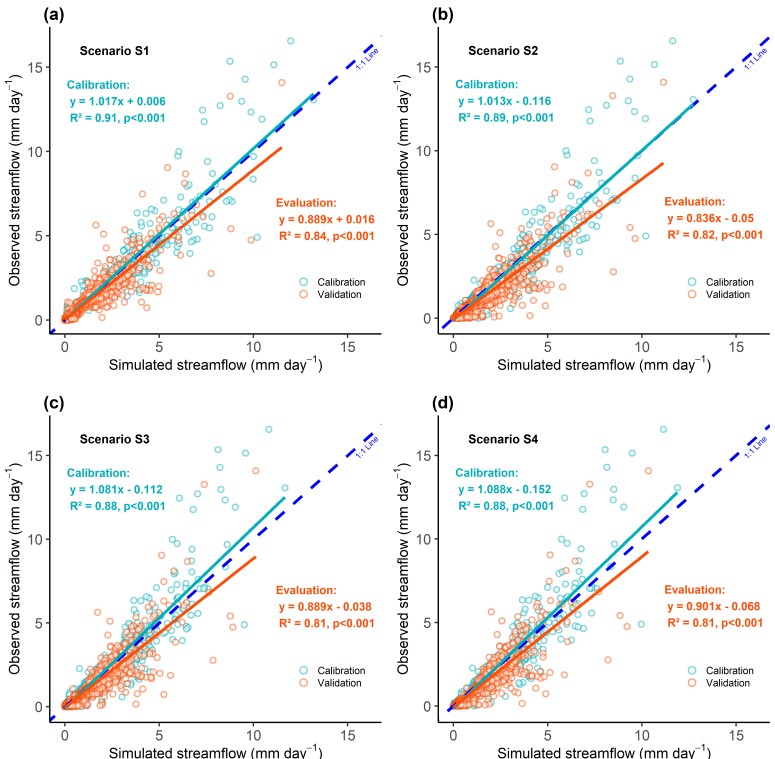

**Figure A1**. Relationship between observed and simulated streamflow for calibration and evaluation periods for four calibration scenarios: a) S1 (Hydro Only), b) S2 (Hydro + DOC), c) S3 (Hydro + $NO_3^-$) and d) S4 (Hydro + DOC + $NO_3^-$). The dashed blue line is the 1:1 line. The light green or orange lines are linear regressions for the calibration or evaluation period, respectively. All relationships were statistically significant ($p < 0.001$).

## Appendix B: Concentration-discharge relationship

In general, the concentration-discharge (CQ) relationship allows three export regimes to be distinguished: (i) chemodynamic with an accretion pattern, (ii) chemodynamic with a dilution pattern or (iii) chemostatic (Godsey et al., 2009; Musolff et al., 2017; Winter et al., 2021). "Chemodynamic" means that the variability in a solute's concentration is similar to or higher than that in Q, with concentrations either increasing (accretion) or decreasing (dilution) as Q increases (Winter et al., 2021). In contrast, chemostatic regimes have constant in-stream nutrient concentrations that are not significantly correlated with Q and have much lower variability (Bieroza et al., 2018). The slope of the linear relationship between ln(C) and ln(Q) (CQ-slope) determines the export regime: (i) chemodynamic with an accretion pattern when greater than 0.1, (ii) chemodynamic with a dilution pattern when less than −0.1 and (iii) chemostatic from −0.1 to 0.1 (Winter et al., 2021). The thresholds of -0.1 and 0.1 for the chemostatic regime were chosen according to Bieroza et al. (2018) and Winter et al. (2021).

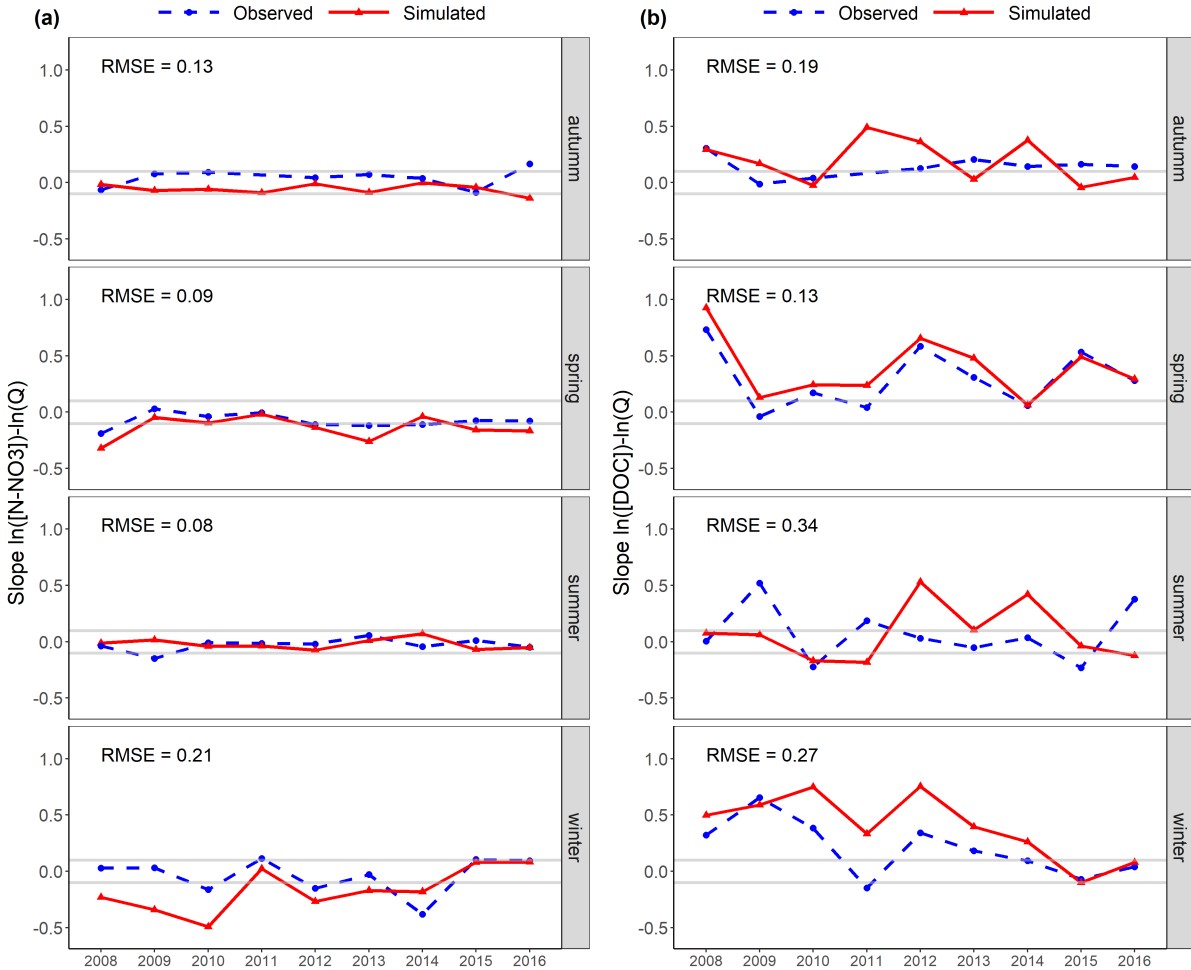

**Figure B1**. a) Slope ln([N-NO$_3$])-ln(Q) for simulated nitrate (NO$_3^-$) data from scenario S3 (streamflow and stream NO$_3^-$ concentration used for calibration). b) Slope ln([DOC])-ln(Q) for simulated DOC data from scenario S2 (streamflow and stream DOC concentration used for calibration). Horizontal grey lines identify the boundary between a chemodynamic regime with a dilution pattern and a chemostatic regime (-0.1) and that between a chemostatic regime and a chemodynamic regime with an accretion pattern (0.1). RMSE: Root-mean-square error.

**Appendix C: Performance criteria**


To evaluate model performance, the study used the following criteria:

1) Nash-Sutcliffe efficiency (NSE) (Nash and Sutcliffe, 1970):

$$NSE = 1 - \frac{\sum_{i=1}^{n}(Y_{i,obs} - Y_{i,sim})^2}{\sum_{i=1}^{n}(Y_{i,obs} - \overline{Y_{obs}})^2} \tag{1}$$

where $Y_{i,sim}$ is the model output, $Y_{i,obs}$ is the observed value of variable Y for time step i, $\overline{Y_{obs}}$ is the mean value of observation data for the study period and n is the length of the time series. NSE ranges from $-\infty$ to 1, with NSE = 1 being the optimal value if the simulation represents the observations perfectly (Moriasi et al., 2007). NSE describes the variance in the observed values over time that is explained by the model. Negative NSE indicates that model predictions are worse than the mean of all observations. The NSE of the flows ($NSE_Q$) and the NSE of

the logarithm of the flows ($NSE_{logQ}$) evaluated the model's ability to reproduce high flows and low flows, respectively (Gharari et al., 2014).

2) Kling-Gupta model efficiency (KGE) (Gupta et al., 2009):

$$KGE = 1 - \sqrt{(r-1)^2 + (\beta-1)^2 + (\alpha-1)^2} \,, \beta = \frac{\mu_{sim}}{\mu_{obs}} \,, \alpha = \frac{\sigma_{sim}}{\sigma_{obs}} \tag{2}$$

where r is the correlation coefficient, $\beta$ and $\alpha$ are the bias and variability ratio, respectively, between simulations and observations, $\mu$ and $\sigma$ are the mean and standard deviation of the variable, respectively, and indices sim and obs represent simulations and observations, respectively.

The closer the KGE is to 1, the better the model performs, and KGE = 1 expresses a perfect fit between predictions and observations. KGE of 0.70-0.82 is considered average to slightly good model performance, while KGE > 0.82 is considered good to very good (Crochemore et al., 2015).

3) The flow duration curve (FDC), which is the distribution of probabilities of streamflow being greater than or

equal to a given magnitude (Sawicz et al., 2011). In the present study, the NSE of the FDC ($NSE_{FDC}$) evaluated the model's ability to reproduce FDCs:

$$NSE_{FDC} = 1 - \frac{\sum_{j=0}^{100}(FDC_{j,obs} - FDC_{j,sim})^2}{\sum_{j=0}^{100}(FDC_{i,obs} - \overline{FDC_{obs}})^2} \tag{3}$$

where $FDC_{j,obs}$ is the FDC of the observed discharge with j probability of exceedance, $FDC_{j,sim}$ is the FDC of the simulated discharge with j probability of exceedance and $\overline{FDC_{obs}}$ is the mean observed discharge (Euser et al., 2013).

4) Volumetric efficiency (VE):


$$VE = 1 - \frac{\sum_{i=1}^{n}|Q_{i,sim} - Q_{i,obs}|}{\sum_{i=1}^{n} Q_{i,obs}} \tag{4}$$

where $Q_{i,obs}$ and $Q_{i,sim}$ are the observed and simulated discharge, respectively, at time step i. VE thus ranges from 0-1 and represents the fraction of simulated water delivered at the correct time (Criss and Winston, 2008).

5) Runoff (RUNOFF [-]), which equals long-term mean streamflow (Q) divided by long-term mean precipitation (P) (Sawicz et al., 2011):

$$RUNOFF = \frac{Q}{P} \tag{5}$$

RUNOFF represents the long-term water balance between water released from the catchment as streamflow and as evapotranspiration (assuming no net change in storage). A high or low runoff ratio indicates a large amount of water exiting as streamflow (dominated by streamflow or blue water) or evapotranspiration (dominated by evapotranspiration or green water), respectively (Sawicz et al., 2011). $NSE_{RUNOFF}$ corresponds to the NSE with RUNOFF as the variable.


6) Root-mean-square error (RMSE):

$$RMSE = \sqrt{\frac{1}{n}\sum_{i=1}^{n}\left(Y_{i,obs} - Y_{i,sim}\right)^2} \tag{6}$$

where $Y_{i,sim}$ and $Y_{i,obs}$ are the simulated and observed value of variable Y, respectively, at time step i and n is the length of the time series. RMSE is easy to interpret because it uses the same units as the model output. The lower the RMSE, the better the model performance.

7) Percentage bias (PBIAS) (Moriasi et al., 2007):

$$PBIAS = \frac{1}{n}\sum_{i=1}^{n}\left(\frac{Y_{i,sim}-Y_{i,obs}}{Y_{i,obs}}\right)*100 \tag{7}$$

measures the mean difference between observations $Y_{i,obs}$ and model simulations $Y_{i,sim}$ of variable Y at time step i. n is the length of the time series. The optimal value of PBIAS is 0.0, with small values indicating accurate prediction and larger positive or negative values indicating overprediction or underprediction bias, respectively.

8) Coefficient of determination (R²):

$$R^2 = \frac{\left(\sum_{i=1}^{n}(Y_{i,obs}-\overline{Y_{obs}})\cdot(Y_{i,sim}-\overline{Y_{sim}})\right)^2}{\sum_{i=1}^{n}(Y_{i,obs}-\overline{Y_{obs}})^2\cdot\sum_{i=1}^{n}(Y_{i,sim}-\overline{Y_{sim}})^2} \tag{8}$$

where $Y_{i,sim}$ and $Y_{i,obs}$ are the simulated and observed value of variable Y, respectively, at time step i, $\overline{Y_{obs}}$ and $\overline{Y_{sim}}$ are the mean value for observed and simulated data for the study period, respectively, and n is the length of the time series. $R^2$ evaluates how accurately the model predicts the variation in observed values. It can reveal the strength and direction of a linear relation between predictions and observations.

*Data availability*. The weather data are available from the INRAE CLIMATIK platform (https://agroclim.inrae.fr/climatik/, in French). The hydrochemical data (i.e. streamflow, groundwater levels, soil water content and solute concentrations) are available from the Observatoire de Recherche en Environnement sur les Agro-Hydrosystèmes (ORE AgrHyS) platform (https://www6.inra.fr/ore_agrhys_eng/Data). ORE AgrHyS, funded by INRAE, is part of the OZCAR French Research Infrastructure (https://www.ozcar-ri.org/agrhys-observatory/).

*Code availability*. The model code is available from https://doi.org/10.5281/zenodo.10161243 or directly from the first author.

*Author contributions*. All co-authors were involved in identifying the research questions, conceptualizing the original methods and interpreting and discussing the results. JSM implemented the model, performed the simulations, created the figures and prepared the first draft of the manuscript. All co-authors contributed to the content and improvement of the manuscript.

*Competing interests*. At least one of the co-authors is a member of the editorial board of *Hydrology and Earth System Sciences*.

*Acknowledgements*. We gratefully acknowledge Chantal Gascuel-Odoux for her insightful comments and suggestions. We thank Yannick Hamon and Mikael Faucheux for conducting field and laboratory work, which provided essential data for this study. We also thank Michelle and Michael Corson for their English and scientific review. This study was performed with the support of the high-performance computing platform MESO@LR at the University of Montpellier. We also thank Jan Seibert and the two anonymous reviewers for their very constructive comments, which helped us improve the manuscript greatly.

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
