# Peer review of "Improving the hydrological consistency of a process-based solute-transport model by simultaneous calibration of streamflow and stream concentrations"

_Hydrology and Earth System Sciences, 2023_

## Author Response (AR1)

Jordy Salmon-Monviola
INRAE - UMR SAS
CS 84215
35042 Rennes Cedex

**Subject:** Submission of a revised version of a manuscript

Rennes, July 8th, 2024.

Dear editor and reviewers,

Please find enclosed a revised version of our manuscript, "Improving the hydrological consistency of a process-based solute-transport model by simultaneous calibration of streamflow and stream concentrations", written by Jordy Salmon-Monviola, Ophélie Fovet and Markus Hrachowitz.

We describe below how we have responded to the issues raised by the reviewers.

Thank you for considering the revised version of our manuscript.

Sincerely,

Jordy Salmon-Monviola on behalf of the co-authors

**Referee comments on "Improving the internal hydrological consistency of a process-based solute-transport model by simultaneous calibration of streamflow and stream concentrations" Salmon-Monviola, J., Fovet, O., and Hrachowitz, M., Hydrol. Earth Syst. Sci. Discuss. https://doi.org/10.5194/hess-2023-292, 2024.**

Referee comments are shown in black. *Authors replies are in blue italic.* Changes in manuscript are in red.

**RC1: 'Comment on hess-2023-292', Anonymous Referee #1, 19 Feb 2024**

In their manuscript, Salmon-Monviola et al. explore the utilization of dissolved organic carbon (DOC) and nitrate concentrations as constraints to refine streamflow predictions and enhance the internal consistency of a conceptual hydrological model. While their investigation revealed that DOC and nitrate concentrations did not enhance streamflow predictions per se, they did, however, reduce the uncertainty in model parameters and the representation of internal hydrological states and flow. The manuscript is notably well-written and clearly makes a case for improving the internal consistency of the conceptual hydrological models by adding additional constraints, such as solute concentrations. It convincingly illustrates how the inclusion of DOC and nitrate concentrations considerably affects the representation of underlying hydrological states and flow. Nonetheless, I have some doubts about whether the constrained version reflects greater realism, as it improved the representation of groundwater storage but showed equal to even worse results for soil moisture (more details in the general comments below). With these concerns addressed, the manuscript holds promise as a significant contribution to the readership of HESS.

*Reply: We thank the reviewer for the positive and constructive feedback that will help us to further improve our work. Below, we outline how we consider responding to the issues pointed out by the reviewer in the revision and which changes we intend to implement.*

**General comments:**

I find it convincing that the simulations, constrained by nitrate concentrations (S3 and S4), have improved the representation of groundwater levels. However, I cannot entirely follow the interpretation of an improved representation of upslope soil moisture. The authors reference Figure 11 to support this claim, yet upon examination, I observe only marginal disparities between the non-constraint simulation (S1) and the one constrained by DOC (S2), while S3 and S4 exhibit notably lower performance in terms of NSE, KGE, PBIAS, and RSME. Consequently, from my perspective, none of the simulations incorporating DOC and/or nitrate concentrations (S2 – S4) consistently elevate internal model consistency in representing soil moisture and groundwater. Hence, it remains uncertain whether these simulations merely exert a general influence on the representation of hydrological states and flows or, indeed, foster an overarching enhancement in the models' internal consistency. It might be that the representation of DOC and nitrate processing and transport are too simple, or soil moisture measurements are not presentative for the entire catchment, as nicely discussed in chapters 4.2 and 4.3. Nevertheless, I find this point insufficiently addressed.

*Reply: We agree with this comment. We will add to the discussion that none of the simulations that include DOC and/or nitrate concentrations (S2–S4) consistently increase the internal model consistency in representing both soil moisture and groundwater.*

The section 3.4.2 has been modified. U-test has been used to show S2 reproduced soil moisture in the upslope zone significantly (p<0.05) better than S1 did (NSE = 0.94 and 0.92, respectively). Section 4.1 has been modified to state that "The improvement in the representation of groundwater level is significant and relatively large for scenarios S2, S3 and

S4 compared to S1 (Fig. 9). In contrast the improvement in the representation of soil moisture in the upslope zone is significant but relatively more limited, only for S2 compared with S1 (Fig. 10b). As a result, only the S2 scenario improves the representation of both groundwater and soil moisture in upslope.".

Elements of discussion have been added in section 4.1 regarding the reasons why from these results, it remains difficult to determine whether including solutes with streamflow in the calibration merely exert only an improvement on the representation of hydrological states and flows of specific storages or whether it actually promotes an overall improvement in the models' internal consistency. There are several reasons, related to the limits of our approach.

**Given general comments of referees we propose to tone down the title of the paper to:**

"Improving the  hydrological consistency of a process-based solute-transport model by simultaneous calibration of streamflow and stream concentrations"

The title can be eventually more toned down:

"Simultaneous calibration of streamflow and stream concentrations in a process-based solute-transport model improves accuracy of model results"

I missed a discussion on the applicability across different catchments, especially in the view that dominant sources and pathways of DOC and nitrate concentrations can strongly vary in other settings. Based on my interpretation of Figures 5 and 6, it seems that DOC concentrations exhibit an enrichment pattern (i.e., increasing concentrations with increasing streamflow), whereas nitrate concentrations demonstrate dilution patterns (i.e., decreasing concentration with increasing discharge). However, these patterns may differ significantly in other catchments with distinct sources and pathways (e.g., Winter et al., 2021; Knapp et al., 2020). How might these differences affect your model setup? From my perspective, it is crucial to address the implications of the specific catchment characteristics and the transferability of your findings to other catchments.

**Reply:** We agree with this comment. We will extend the discussion to address the implications of the specific catchment characteristics and the transferability of our findings to other catchments. A figure (similar to the Figure RC.1) will be added to demonstrate the model's ability to represent different patterns of DOC and nitrate concentrations (dilution, enrichment, and chemostatic) by examining the seasonal evolution of the C[DOC or N]/Q slope throughout the simulation period. The argument is that if the model can replicate multiple patterns, rather than just one characteristic of the Kervidy catchment for each solute, then it may be able to replicate C-Q behaviour in other catchments. It should also be added that the structure of the model is sufficiently flexible to adapt to other catchment characteristics with different DOC and N dynamics than Kervidy. We will consider the suggested articles in the discussion.

In the Figure RC.1 below, the relationship $\log Q = a* \log C + b$ (Knapp et al., 2020) has been calculated for each of the four seasons for the period 2008-2016 for the observed and simulated data for streamflow and solute concentrations (nitrate and DOC). The parameter a represents the relationship between $\ln(C)$ and $\ln(Q)$ ($CQ$ slope), which enables a differentiation between export regimes: (i) chemodynamic with an accretion pattern (a > 0.1), (ii) chemodynamic with a dilution pattern (a < −0.1), and (iii) chemostatic (−0.1 < a < 0.1) (Winter et al., 2021).

The model reproduced relatively well export regime pattern for nitrate and DOC. Performance of the model was slightly lower for DOC than for the nitrate export regime. The model simulated an accretion type export regime for the DOC for some years in autumn (2011 and 2014) and in summer (2012 and 2014), whereas the observed pattern is more chemostatic. As

the hydrological dynamics are simulated relatively well during these periods (Fig. 4), it can be assumed the model should produce too much DOC during these periods. A sensitivity analysis of DOC production to input factors (temperature, soil moisture, reservoir volume) and an analysis of the dynamics of the factors could be carried out to see if these factors explain these model results for DOC in autumn and summer.

[Figure]

Figure RC.1 a) Slope [N-NO3]-Q. Simulated nitrate data from scenario S3 (streamflow and stream NO3 concentration used for calibration) are used. b) Slope [DOC]-Q. Simulated DOC data from scenario S2 (streamflow and stream DOC concentration used for calibration) are used. Horizontal gray lines delineate the boundary between dilution pattern and chemostatic transport (-0.1), and between chemostatic and chemodynamic with an accretion pattern transport (0.1). RMSE: Root-mean-square error.

These elements have been added in the second paragraph of 4.4 section.

**Specific comments:**

Line 95 - 96: What exactly do you mean by substantially differing? Is there a directional relationship between concentrations and discharge, or are dynamics completely independent?

**Reply:** It will be clarified in the manuscript. The point of this sentence is not to make a link between concentration and discharge, but to highlight the fact that the dynamics of solute

concentrations (N, DOC) can be very different between certain hydrological compartments in the catchment and the outlet of a catchment (cf. Aubert et al., 2013; Strohmenger et al., 2020).

Added in Introduction section:

"As the movement of water and solutes through the landscape is inherently coupled (Knapp et al., 2020), using time series of multiple elements along with streamflow during calibration may provide additional insights into the flow paths of water through the catchment (Strohmenger et al., 2021). This potential may be particularly high when using solutes that differ in their sources and flow paths across spatial and temporal scales in a catchment. Calibration that includes streamflow along with solutes that have distinct dynamics, as frequently observed with dissolved organic carbon (DOC) and NO3- (Inamdar and Mitchell, 2006; Taylor and Townsend, 2010), such as in headwater agricultural catchments (Aubert et al., 2013; Strohmenger et al., 2020; Thomas et al., 2016), thus has high potential to constrain models to adequately reproduce water storage dynamics and flow paths"

Line 98: I recommend toning down a little to "can" be closely related.

**Reply:** It will be changed in the manuscript. It has been removed

Line 98 – 104: See here my second general comment. There are various patterns of DOC nitrate export dynamics depending on the storage and flow paths within the catchment. This is a little too simplified for my taste.

**Reply**: The dynamics of nitrate and DOC identified in this catchment and based on previous work (Morel et al., 2009; Aubert et al., 2013; Lambert et al., 2013, 2014; Humbert et al. 2015; Strohmenger et al., 2020) will be detailed in the revised manuscript.

It has been modified, in Introduction section, to state more simply that "Calibration that includes streamflow along with solutes that have distinct dynamics, as frequently observed with dissolved organic carbon (DOC) and NO3- (Inamdar and Mitchell, 2006; Taylor and Townsend, 2010), such as in headwater agricultural catchments (Aubert et al., 2013; Strohmenger et al., 2020; Thomas et al., 2016), thus has high potential to constrain models to adequately reproduce water storage dynamics and flow paths".

The pattern of export dynamics of DOC and nitrate specific to the Kervidy catchment has been described in 2.1 section.

Line 105 – 108: Nice and clear!

**Reply:** Thank you for your comment

Line: 116: Could you write the full name of AgrHyS once, please?

**Reply:** The full name of AgrHyS : AgroHydroSystem, will be added in the manuscript

Added in 2.1. section.

Line 135: Why only in riparian-zone soils and not in all soils? What dynamic do you infer if speaking of unlimited supply? Chemostasis? Enrichment? Information about DOC sources and the relationship between Q and DOC, and Q and nitrate (for example in the SI) could help to back up this argument.

**Reply:** Using end-member mixing analysis to identify DOC sources and quantify their respective contribution to the DOC stream in the agricultural headwater catchment of Kervidy-Naizin (France), Morel et al. (2009) calculated that between 64 and 86% of the DOC that entered the stream during storms, when much of the DOC export from soils to streams and rivers occurs (Lambert al., 2014), originated from riparian wetland soil. This result confirms previous studies showing that the riparian soils are the main source of DOC in most headwater catchments (Lambert al., 2013). Morel et al. (2009) also demonstrates that this riparian wetland zone in Kervidy-Naizin behaved as a non-limiting storage of DOC during the flushing process. Hillslope soils were also found to contribute to stream DOC export. However, changes in dissolved organic matter (DOM) composition determined by isotopic and spectroscopic analyses revealed that DOM stored in the upland soils were supply-limited and thus was seasonally depleted after the rise of groundwater in these areas (Lambert et al., 2013). Lambert et al. (2014) determined that upland DOC contribution decreased from ca. 30% of stream DOC flux at the beginning of the high-flow season to <10% later in the season in the Kervidy-Naizin catchment.
We will add a summary of these elements and a figure in SI showing the seasonal variability of the C-Q relationship for DOC and NO3 (simulated and observed, cf. previous reply to general comment).

Added in 2.1 section

Line 379: Those deep infiltration losses appear a little bit like a "mathematical marionette" to me, and they appear to be a highly sensitive parameter. Can you please elaborate on those a little more?

**Reply**: The deep infiltration losses parameter is used in this study to explicitly account for intercatchment groundwater flows defined as groundwater fluxes crossing topographic divides, implying that precipitation falling in one catchment affects the streamflow in another catchment (Bouaziz et al., 2018).
Analysis of the long-term water balance in a similar physiographic headwater catchment in Brittany revealed a significant deficit (Hrachowitz et al., 2014). Although this cannot be fully verified, as pointed out by Beven (2001), there is evidence that such deficits are in many catchments- at least partly-caused by significant intercatchment groundwater flow (Le Moine et al., 2007; Schaller and Fan, 2009, Hrachowitz et al., 2014). In addition, data from 58 catchments in the Meuse basin provide evidence of the likely presence of significant net intercatchment groundwater flows occurring mainly in small headwater catchments underlain by fractured aquifers (Bouaziz et al., 2018) such as the Kervidy catchment. Indeed, this catchment consists of a variety of Brioverian schists of low permeability and lies below a fissured and fractured weathered layer of variable thickness 1-30 m deep (Molenat et al., 2005). The use of the deep infiltration losses parameter is also to allow both the reproduction during the summer period of zero flow at the outlet and the groundwater dynamics with a long recession observed, regardless of the piezometer where it can be recorded (see Fig. 9 or Humbert et al., 2015).
Taking these different arguments into account, we found it preferable to explicitly represent inter-catchment groundwater flows in the modelling of the Kervidy catchment. Such deep infiltration losses are rarely considered in standard formulations of common conceptual models. A few conceptual models explicitly account for net intercatchment groundwater flows such as the GR4J empirical model (Perrin et al., 2003) often applied in French catchments, HYDROLOG (Chiew and McMahon, 1990), SMAR (Goswami and O'Connor, 2010), mHM

(Samaniego et al., 2011) and the flexible model structure used in Hrachowitz et al. (2014). Neglecting intercatchment groundwater flows in conceptual models may still result in high performances of streamflow simulation; however, may introduce misrepresentation of the natural system in hydrological models in particular an overestimating actual evaporation rates to compensate for this lack (Le Moine, 2008; Hrachowitz et al., 2014). For our study, in the absence of detailed process knowledge, the deep infiltration losses from the study catchment have been conceptualized as a loss term in the slow groundwater reservoir. The results of the different calibration scenarios showed the importance of this parameter in ensuring that the model reproduced the observed groundwater dynamics, particularly in scenario S4 (Fig. 9). In this scenario, where all constraints (streamflow, DOC and nitrate concentration) are considered, the distribution of this parameter differed from that of the other scenarios, with higher values (Fig. 8j). On the basis of these results, it can be assumed that for the model to best reproduce the dynamics of streamflow, concentrations (DOC, nitrate) and groundwater, it is necessary to represent an intercatchment groundwater flow of the order of 0.35 mm. $day^{-1}$ (best compromise parameter value for S4).

We will add some of these explanatory notes on the deep infiltration losses parameter in section 2.3.1 of the revised manuscript.

Added in 2.3.1 section.

Line 454 – 457: This sentence is very hard to read. Can you rewrite or slit it, please?

**Reply:** This sentence will be reworded to make it clearer and more concise.

The sentence has been split.

Line 527 - 529: See my first major comment. The improvement in groundwater storage is convincing, but I do not see a significant improvement in upslope soil moisture. Here, S1 performs similarly well to S2 and clearly better than S3 and S4. I find it critical that S1 seems to perform better for soil moisture, while S3 and S4 perform better for groundwater. Thus, neither DOC nor NO3 seems to consistently improve the internal representation of water fluxes.

**Reply:** We agree with this comment. We will add to the discussion to that none of the simulations that include DOC and/or nitrate concentrations (S2 – S4) consistently increase the internal model consistency in representing both soil moisture and groundwater.

Section 4.1 has been modified to state that "The improvement in the representation of groundwater level was significant and relatively large for scenarios S2, S3 and S4 compared to S1 (Fig. 9). In contrast, the improvement in the representation of soil moisture in the upslope zone was significant but relatively small only for scenario S2 compared to S1 (Fig. 10b). Thus, only scenario S2 improved the representation of both groundwater and soil moisture in the upslope zone.".

Line 541 – 543: Consequently, this appears overstated to me. I only see that it improves the groundwater storage representation, while two out of three scenarios show a lower performance for soil moisture.

**Reply:** We will add to the discussion to that none of the simulations that include DOC and/or nitrate concentrations (S2–S4) consistently increase the internal model consistency in representing both soil moisture and groundwater.

Line 556 – 575: I agree with all points discussed here, but I am not entirely convinced that adding DOC and NO3 made the model produce the 'right answers for the right reason' for the reasons mentioned above.

**Reply:** We will add sentences in the discussion section to clarify that we have indeed added solutes here (NO3 and DOC) that have opposite dynamics (Stromhenger et al., 2020) and for which conceptual formalisms have been successfully tested in the literature (Birkel et al., 2014; Fovet et al., 2015), with the aim of adding a useful constraint to the hydrological modelling. However, our results show that none of the simulations including DOC and/or nitrate concentrations (S2-S4) consistently increase the internal model consistency in the representation of both soil moisture and groundwater, and thus our approach does not fully produce the 'right answers for the right reason'.

Our results depend on the assumed dominant hydrological and biogeochemical processes, implemented in the model. Such an approach is limited by our incomplete knowledge of the catchment and the processes underlying the system response. In addition, the representation of reactive solutes increases the number of parameters and the complexity of the approach. It would be interesting to compare our approach with the introduction of non-reactive solutes, such as stable isotopes of water to see if this can better reproduce the dynamics of both the soil moisture and groundwater.

Added in the third paragraph of 4.1 section.

Line 602 – 604: Good point. I also enjoyed reading chapter 4.2 and 4.3.

**Reply:** Thank you for your comment

Line 667 - 672: This appears contradictory to me. Did it or did it not improve the models' ability to reproduce streamflow? The first sentence says no, the second yes.

**Reply:** This sentence will be reworded to be clearer and more concise.

These elements have been clarified in Conclusion section by considering on the one hand the best compromise solution and on the other hand by considering all the hydrological signatures for discharge obtained from discharge:

"Given the performance of the best-compromise model for each scenario, the results of this study tend to reject the first hypothesis, as using daily stream DOC and NO3- concentrations along with streamflow data to calibrate the model did not improve the model's performance for simulated streamflow for the calibration or evaluation period compared to calibration with streamflow alone. In contrast, considering all hydrological signatures for discharge obtained from the envelope, the scenario that included NO3- along with streamflow improved the model's ability to reproduce streamflow, especially low flows."

**Figures**

Figure 1: Why show the entire Narzin catchment if your results focus on the Kervidy-Narzin catchment only? In the caption, you do not mention the Narzin catchment either – It seems more straightforward to me to show the Kervidy-Narzin catchment only.

**Reply:** Our results focus on the Kervidy catchment. However, it was important to show that the Kervidy catchment is a sub-catchment of the Naizin catchment by illustrating their respective boundaries. We used data from the weather station and the Toullo station, situated beyond the Kervidy boundary but within the Naizin catchment. By showing that these two catchments are nested, we strengthen the hypothesis that, although the Toullo station is located outside the Kervidy catchment, it can represent the catchment's soil moisture conditions in the upland zone of Kervidy. We will mention the Naizin catchment more explicitly in the figure caption.

Added in caption of Fig. 1 and in 2.2 section.

Figures 4, 5, and 6: Adding the scenario names (S1 – S4) to the figures would improve clarity compared to only mentioning them in the description below. Moreover, I find it difficult to see the differences between the model runs. You might consider adding an observed vs. simulated plot with a 1:1 line to the SI. But this is just a suggestion.

**Reply**: Thanks for the suggestion. We will add the scenario names in figures 4 to 6. An observed vs. simulated plot with a 1:1 line will be added either inside each figure or in the SI.

The scenarios names have been added in Fig. 4. Observations have been represented with dots rather than lines. An observed vs. simulated plot with a 1:1 line has been added to the SI considering calibration and evaluation periods. In the 3.2 part we have added: "Overall, model performances for the evaluation period were only slightly lower than those for the calibration period for all four scenarios (Figs.4 and A1)."

Figure 9: consider using a white font in front of the dark blue box for S1. A black font is hard to read with dark background (this also applies to the other figures).

**Reply**: We will try to make the table in this figure clearer.

The boxes in Figure 9 has been modified

Figures 9, 10, and 11 could be combined and reduced in height to save space.

**Reply**: We will try to reduce the height of these figures or move some of them to the SI.

Figures have been reduced in height. Figure 10 et 11 have been combined.

Overall, I do think 12 Figures, each of them taking up around half a page, is too much. This might be a matter of taste, but I would recommend combining figures and lowering their height and/or shifting some of them to the SI.

**Reply**: Thank you for your suggestion. We will try to reduce the height of the figures or move them to the SI. A table with metrics for hydrology and concentrations summarizing the results instead of a figure could be a solution.

The figures 5, 6, 8, 11 have been reduce in size.

Knapp, J. L., Freyberg, J. von, Studer, B., Kiewiet, L., and Kirchner, J. W.: Concentration-discharge relationships vary among hydrological events, reflecting differences in event characteristics, Hydrol. Earth Syst. Sci. Discuss., 1–27, https://doi.org/10.5194/hess-24-2561-2020, 2020.

Winter, C., Lutz, S. R., Musolff, A., Kumar, R., Weber, M., and Fleckenstein, J. H.: Disentangling the impact of catchment heterogeneity on nitrate export dynamics from event to long-term time scales, Water Resour. Res., 57, e2020WR027992, https://doi.org/10.1029/2020WR027992, 2021.

**Reply**: These references will be added in the revised manuscript

Added

References:

Aubert, A. H., Gascuel-Odoux, C., Gruau, G., Akkal, N., Faucheux, M., Fauvel, Y., Grimaldi, C., Hamon, Y., Jaffrézic, A., Lecoz-Boutnik, M., Molénat, J., Petitjean, P., Ruiz, L., and Merot, P.: Solute transport dynamics in small, shallow groundwater-dominated agricultural catchments: insights from a high-frequency, multisolute 10 yr long monitoring study, Hydrol. Earth Syst. Sci., 17, 1379–1391, 2013.

Beven, K.: On hypothesis testing in hydrology, Hydrol. Process., 15, 1655–1657, 2001.

Birkel, C., Soulsby, C., and Tetzlaff, D.: Integrating parsimonious models of hydrological connectivity and soil biogeochemistry to simulate stream DOC dynamics: PARSIMONIOUS COUPLED DOC MODEL, J. Geophys. Res. Biogeosci., 119, 1030–1047, 2014.

Bouaziz, L., Weerts, A., Schellekens, J., Sprokkereef, E., Stam, J., Savenije, H., and Hrachowitz, M.: Redressing the balance: quantifying net intercatchment groundwater flows, Hydrol. Earth Syst. Sci., 22, 6415–6434, https://doi.org/10.5194/hess-22-6415-2018, 2018.

Chiew, F. and McMahon, T.: Estimating groundwater recharge using a surface watershed modelling approach, J. Hydrol., 114, 285–304, https://doi.org/10.1016/0022-1694(90)90062-3, 1990.

Fovet, O., Ruiz, L., Faucheux, M., Molénat, J., Sekhar, M., Vertès, F., Aquilina, L., Gascuel-Odoux, C., and Durand, P.: Using long time series of agricultural-derived nitrates for estimating catchment transit times, Journal 855 of Hydrology, 522, 603–617, 2015.

Goswami, M. and O'Connor, K. M.: A "monster" that made the SMAR conceptual model "right for the wrong reasons", Hydrolog. Sci. J., 55, 913–927, 2010. https://doi.org/10.1080/02626667.2010.505170

Hrachowitz, M., Fovet, O., Ruiz, L., Euser, T., Gharari, S., Nijzink, R., Freer, J., Savenije, H. H. G., and Gascuel-Odoux, C.: Process consistency in models: The importance of system signatures, expert knowledge, and process complexity, Water Resour. Res., 50, 7445–7469, 2014. https://doi.org/10.1002/2014WR015484

Humbert, G., Jaffrezic, A., Fovet, O., Gruau, G., and Durand, P.: Dryseason length and runoff control annual variability in stream DOC dynamics in a small, shallow groundwater-dominated agricultural watershed, Water Resour. Res., 51, 7860–7877, 2015. doi:10.1002/2015WR017336.

Lambert, T., Pierson-Wickmann, A. C., Gruau, G., Jaffrezic, A., Petitjean, P., Thibault, J. N., and L. Jeanneau: Hydrologically driven seasonal changes in the sources and production mechanisms of dissolved organic carbon in a small lowland catchment, Water Resour. Res., 49, 5792–5803, 2013. doi:10.1002/wrcr.20466.

Lambert, T., Pierson-Wickmann, A. C., Gruau, G., Jaffrezic, A., Petitjean, P., Thibault, J. N., and L. Jeanneau: DOC sources and DOC transport pathways in a small headwater catchment as revealed by carbon isotope fluctuation during storm events, Biogeosciences, 11(11), 3043–3056, 2014.

Le Moine, N.: Le bassin versant de surface vu par le souterrain: une voie d'amélioration des performances et du réalisme des modèles pluie-débit ?, PhD thesis, CEMAGREF, UR HBAN, Antony, France, 2008.

Le Moine, N., Andréassian, V., Perrin, C., and Michel, C.: How can rainfall-runoff models handle intercatchment
groundwater flows? Theoretical study based on 1040 French catchments, Water Resour. Res., 43, W06428,
https://doi.org/10.1029/2006WR005608, 2007.

Molenat, J., Gascuel, C., Davy, P., and Durand, P.: How to model shallow water-table depth variations: the case of the Kervidy-Naizin catchment -France, Hydrological Processes, 19, 901–920, https://doi.org/10.1002/hyp.5546, 2005.

Morel, B., Durand, P., Jaffrezic, A., Gruau, G., and Molenat, J.: Sources of dissolved organic carbon during stormflow in a headwater agricultural catchment, Hydrol. Processes, 23(20), 2888–2901, 2009.

Perrin C., Michel C., and Andréassian V.: Improvement of a parsimonious model for streamflow simulation, J Hydrol., 279, 275–289, https://doi.org/10.1016/S0022-1694(03)00225-7, 2003.

Samaniego, L., Kumar, R., and Jackisch, C.: Predictions in a datasparse region using a regionalized grid-based hydrologic model driven by remotely sensed data, Hydrol. Res., 42, 338–355, 2011.

Schaller, M. F. and Fan, Y.: River basins as groundwater exporters and importers: Implications for water cycle and climate modeling, J. Geophys. Res.-Atmos., 114, D04103, https://doi.org/10.1029/2008JD010636., 2009.

Strohmenger, L., Fovet, O., Akkal-Corfini, N., Dupas, R., Durand, P., Faucheux, M., Gruau, G., Hamon, Y., Jaffrezic, A., Minaudo, C., Petitjean, P., and Gascuel-Odoux, C.: Multitemporal Relationships Between the Hydroclimate and Exports of Carbon, Nitrogen, and Phosphorus in a Small Agricultural Watershed, Water Resour. Res., 56, https://doi.org/10.1029/2019WR026323, 2020.

**Referee comments on "Improving the internal hydrological consistency of a process-based solute-transport model by simultaneous calibration of streamflow and stream concentrations" Salmon-Monviola, J., Fovet, O., and Hrachowitz, M., Hydrol. Earth Syst. Sci. Discuss. https://doi.org/10.5194/hess-2023-292, 2024.**

Referee comments are shown in black. *Authors replies are in blue italic.* Changes in manuscript are in red.

**RC2: 'Comment on hess-2023-292', Anonymous Referee #2, 06 Mar 2024**

The study by Salmon-Monviola et al. investigates the utility of using biogeochemistry parameters to improve a conceptual hydrological model's accuracy and internal consistency. The authors modeled hydro-biogeochemical processes in the Kervidy-Naizin catchment in NW France to test this idea. They used a hydrological model based on previously published models, e.g., Hrachowitz et al. (2015). Adding DOC and NO3- processes into the model did not improve streamflow prediction. However, adding solute processes to streamflow improved the model's internal consistency, as demonstrated by the ability to model groundwater level and upslope soil moisture and reduce parameter uncertainty. The paper was generally well organized and well written, and the aims of the paper are within the scope of HESS. The paper requires some revisions, but if done satisfactorily, this paper would be a solid addition to the literature.

*Reply: We thank the reviewer for the positive and constructive feedback that will help us to further improve our work. Below, we outline how we consider responding to the issues pointed out by the reviewer in the revision and which changes we intend to implement.*

**General comments:**

I agree with Reviewer 1's 'General Comments' section. Figure 11 does not provide strong enough evidence to suggest that adding DOC to the model (S2) improves model (S1) accuracy. I would also like to echo the need for discussion on the applicability of these findings across regions.

*Reply: We thank the two reviewers for their general comments. The discussion section will be revised to address two points. The first concerns the fact that the inclusion of DOC in the model does not provide sufficiently robust evidence to show that DOC provides a better representation of soil moisture in the upper slopes. The other concerns the transferability of our results to other catchments. We will also focus more in the manuscript on the model's ability to reproduce streamflow and solutes (NO3, DOC) concentrations simultaneously.*

The section 3.4.2 has been modified. U-test has been used to show S2 reproduced soil moisture in the upslope zone significantly ($p<0.05$) better than S1 did (NSE = 0.94 and 0.92, respectively). Section 4.1 has been modified to state that "The improvement in the representation of groundwater level is significant and relatively large for scenarios S2, S3 and S4 compared to S1 (Fig. 9). In contrast the improvement in the representation of soil moisture in the upslope zone is significant but relatively more limited, only for S2 compared with S1 (Fig. 10b). As a result, only the S2 scenario improves the representation of both groundwater and soil moisture in upslope.".

Elements of discussion have been added in section 4.1 regarding the reasons why from these results, it remains difficult to determine whether including solutes with streamflow in the calibration merely exert only an improvement on the representation of hydrological states and flows of specific storages or whether it actually promotes an overall improvement in the models' internal consistency. There are several reasons, related to the limits of our approach.

**Given general comments of referees we propose to tone down the title of the paper to:**

"Improving the  hydrological consistency of a process-based solute-transport model by simultaneous calibration of streamflow and stream concentrations"

The title can be eventually more toned down:

"Simultaneous calibration of streamflow and stream concentrations in a process-based solute-transport model improves accuracy of model results"

Discussion on the applicability of these findings across regions has been added in the second paragraph of 4.4 section.

I also found it difficult to follow how results were produced and interpreted. This may be because I don't have experience with this type of modeling. However, HESS has a broad hydrological readership, so the following comments are aimed to help communicate to a broader audience. For example, it wasn't easy to follow how the DOC and NO3 parameters were incorporated into the conceptual model. The hydrology was well outlined in Figure 2, and it would help to have a similar figure (perhaps Fig. 2b) for the biogeochemistry.

**Reply:** Thanks for the suggestion. Figure 2 will be modified to better identify where the NO3 and DOC parameters are incorporated in the model.

Biogeochemistry parameters have been added in the Fig. 2.

Secondly, many model performance metrics (e.g., PBIAS, RMSE, eCDF, etc.) were not adequately explained or introduced, making it challenging to follow model interpretations. Additionally, the results alluded to some statistics (e.g., line 515), but no statistical methods were described.

**Reply:** We agree with the reviewer. Performance metrics will be better explained in the SI. The eCDF is not a performance measure but the empirical cumulative distribution function. It will better explained in Figure 8. The statistical method used to compare the distribution of a variable under different scenarios will be described.

Performance metrics have been more explained in the SI. The non-parametric Mann-Whitney U test was used to test whether model predictions of calibration scenarios S2, S3 and S4 differed significantly ($p < 0.05$) from those of the baseline scenario S1 (added in 5.2 section).

Another inconsistency was in the figures. Figures 4-6 had distinct calibration and evaluation periods for streamflow, DOC, and NO3 concentrations.

**Reply:** Calibration and evaluation periods are not distinct for streamflow, DOC, and NO3 concentrations in Figure 4-6. It is described in 2.5 section. The calibration period was set from 1 Jan 2013 to 1 Sep 2016, while the evaluation period was set from 1 Aug 2008 to 31 Dec 2011 for each scenario.

Yet for figures 9-11 showing groundwater level and soil moisture, the whole period appeared to be for evaluation, but the date ranges on the x-axis varied. It wasn't clear how these modeling approaches differed and why the date ranges changed.

**Reply:** In Section 2.5, we explained that in the later evaluation step, observed soil water content and groundwater level measurements were used as independent data to assess the consistency of the internal processes of the best compromise model for each scenario obtained during the calibration period. This was done by running a simulation over the period 2008-2017 for each

of the best compromise parameter sets of each scenario obtained during the calibration step. In this way, simulated data were compared with observed data for different time periods, as observed data are not available for identical time periods. In section 2.2 we explained that groundwater level data are available since 2000, soil moisture data in the upland zone (Toullo station) are available from 1 Jan 2016 to 1 Jan 2019 and soil moisture data in the riparian zone (PG2) are available from 3 Dec 2013 to 1 Jan 2017. These differences in observed data availability periods explain why the date ranges on the x-axis vary between Figures 9 and 11.

Finally, as a stylistic point, I found all the model abbreviations and acronyms in the text highly distracting. I suggest that to improve readability, the authors should only use common abbreviations (e.g., DOC) in the text.

**Reply:** We understand this comment. However, we consider that using the long names of each variable and parameter in the model would also make the text too cumbersome. All the model's abbreviations and acronyms are defined in table A1 in the SI.

**Specific comments:**

Lines 87-88: vague sentence. The example needs to be more concrete.

**Reply:** It will be clarified in the manuscript.

Added in Introduction section: "Alternately, one may seek to extract more information from the available data, for example by developing signatures that represent different aspects of the data (Euser et al., 2013; Gharari et al., 2014; Fenicia et al., 2018), and then compare the signatures of the observed and simulated time series. For streamflow, the hydrological signatures can include quantiles of the streamflow distribution (values of the flow duration curve (FDC)), the base flow index, the flashiness index and many others (e.g., Kavetski et al., 2018). '

Lines 95-97: What is "This potential" referring to? The spatial distribution of solutes where? In groundwater?

**Reply:** It will be clarified in the manuscript.

It has been clarified in Introduction section:" As the movement of water and solutes through the landscape is inherently coupled (Knapp et al., 2020), using time series of multiple elements along with streamflow during calibration may provide additional insights into the flow paths of water through the catchment (Strohmenger et al., 2021). This potential may be particularly high when using solutes that differ in their sources and flow paths across spatial and temporal scales in a catchment. Calibration that includes streamflow along with solutes that have distinct dynamics, as frequently observed with dissolved organic carbon (DOC) and $NO_3^-$ (Inamdar and Mitchell, 2006; Taylor and Townsend, 2010), such as in headwater agricultural catchments (Aubert et al., 2013; Strohmenger et al., 2020; Thomas et al., 2016), thus has high potential to constrain models to adequately reproduce water storage dynamics and flow paths"

Line 130: are 'livestock units' different from the number of animals?

**Reply:** The concept of a livestock unit (LU) was originally intended to reflect the animal stocking rate on the farm according to the energy requirements of the animals (Benoit and Veysset, 2021). LU is different from the number of animals. The reference 'Benoit and Veysset, 2021' will be added in the manuscript.

The reference (Benoit and Veysset, 2021) has been added in 2.1 section.

Line 157: define TDR

**Reply:** The term TDR **(**Time Domain Reflectometry**)** will be defined in the revised manuscript

Term added in 2.2 section.

Lines 219-223: need more clarification in this section. Are you suggesting more N is removed in winter via denitrification, and in summer by biological uptake?

**Reply:** It will be clarified in the manuscript. The period over which heterotrophic denitrification is most important is not known for the Kervidy catchment, as we have no observations of denitrification rates on this site. Temporal pattern of denitrification generally observed for agricultural headwater is that denitrification rates were initially low coming out of the winter, increased during the spring, peaked in summer, and decreased in the fall before reaching their lowest in the middle of winter (Anderson et al., 2014). In agricultural landscapes where N is abundant and available beyond what is required for plant growth, primary controls on denitrification shift to a dependence on C availability, $O_2$ status, and temperature (Barton et al., 1999). Riparian areas of these systems are often abundant in C. Thus, expect rates will be highest in the late spring-summer-early fall months when temperatures are warmer and $O_2$ lower as long as the soils remain wet (Anderson et al., 2014). However, differences in this seasonal dynamic can be observed and can generally be attributed to the availability of $NO_3$, which reflects the competition between denitrifiers and vegetation, and to groundwater levels, which provide optimal conditions for denitrification (Hefting et al.,2003). We also have no observed data on the biological transformation of nitrate through consumption by aquatic primary producers, although we can suppose that this biological activity is greater in spring and summer.
Thus, in the absence of precise knowledge of the temporal pattern of the nitrate biological removal in the Kervidy catchment, we have chosen to represent the biological removal as a constant in a parsimonious modelling approach. And we considered that if, on the basis of this constant, important nitrate biological removal were simulated in winter compared to the pattern generally observed in agricultural landscapes, the effect on nitrate concentration would be negligible given the high nitrate load in winter in Kerviy-Naizin.

It has been added in 2.3.2 section

Lines 232-233: The two Birkel et al. papers cited here are from catchments with wetlands supplying the bulk of the DOC. So, it isn't surprising that the stream DOC reactivity is negligible. However, DOC is typically more reactive in agricultural catchments (see (Shang et al 2018, Eder et al 2022)). However, it doesn't seem like stream water is a reservoir with an associated transit time in the conceptual model, so how would stream water DOC (and NO3) reactivity be included? It seems like your model is well-positioned to speak to solute groundwater transport, but less so for surface water.

**Reply:**

**Stream water DOC reactivity:**

Using end-member mixing analysis to identify DOC sources and quantify their respective contribution to the DOC stream in the agricultural headwater catchment of Kervidy-Naizin (France), Morel et al. (2009) show that stream DOC dynamics in winter storm events, when much of the DOC export from soils to streams and rivers occurs (Lambert al., 2014), can be explained by catchment processes, with an insignificant role of in-stream sources. This study confirms previous findings that in streams draining headwater catchments, most of the dissolved organic carbon (DOC) is thought to be primarily of external (allochthonous) origin, resulting from the interaction between biogeochemical and hydrological processes in soils

(Lambert al., 2013). They calculated that between 64 and 86% of the DOC that entered the stream during storms originated from riparian wetland topsoil, confirming also previous studies that shown that the riparian soils are the main DOC source in most headwater catchments (Lambert al., 2013). Morel et al. (2009) also demonstrates that this riparian wetland zone in Kervidy-Naizin behaved as a non-limiting store during the flushing process. Hillslope soils were also found to contribute to stream DOC export. However, changes in dissolved organic matter (DOM) composition determined by isotopic and spectroscopic analyses revealed that DOM stored in the upland soils were supply-limited and thus was seasonally depleted after the rise of groundwater in these areas (Lambert et al., 2013). Lambert et al. (2014) determined that upland DOC contribution decreased from ca. 30% of stream DOC flux at the beginning of the high-flow season to <10% later in the season in the Kervidy-Naizin catchment.

While the Kervidy-Naizin catchment is heavily affected by the intense agriculture activities as revealed by the average streamwater nitrate concentration of 70 mg/l, Morel et al. (2009) show that the dynamics of DOC transfer here is not fundamentally different from that observed in alpine or/and forested catchments. Furthermore, we have no observations to suggest that there is any production of autochthonous DOC in the low order Kervidy stream.

This part has been added in 2.1 section.

This is consistent with study of Shang et al. (2018) which demonstrates an increasing contribution of protein-like, autochthonous DOM, accompanied by a decline in percent contribution of allochthonous DOM, from low-order to high-order systems. Such a pattern is consistent with a general conceptual trend describing DOM transformations along fluvial continuum. That is, in-stream biogeochemical processing becomes increasingly important from headwaters to large downstream rivers, due to an increased open-canopied area that stimulates photosynthetic microorganisms as well as a longer residence time allowing more thorough biological processing of DOM (Shang et al., 2018).

Taken together, these results lead us to not represent the reactivity of DOC in stream water in our model and to assume that in-stream processes have a negligible influence on DOC concentrations. This led us to consider the assumptions of Birkel et al. (2014, 2020) as valid in the context of the Kervidy catchment.

We agree that our model is not yet able to represent the recent results of Eder et al (2022) study highlighting instream DOC production in the catchment of The Hydrological Open Air Laboratory (HOAL) Petzenkirchen (western part of Lower Austria, 66 ha).

This part has been added in 2.3.3 section

**Stream water NO3 reactivity:**

Denitrification can be a sink for nitrate in streams, particularly small (low-order) ones (Böhlke et al. 2009). However, methods for measuring in-stream denitrification are difficult and have large uncertainties, and the controlling variables are not known well enough to make reliable predictions for targeted management decisions (Böhlke et al. 2009). Given the lack of in-stream denitrification measurements and the low potential for in-stream nitrate removal (estimated at ca. 4 % per year, Salmon-Monviola et al., 2013) in Kervidy-Naizin, we did not model it and thus assumed zero in-stream denitrification.

A summary of these various elements will be included in the revised manuscript.
It has been added in 2.3.2 section

Lines 235-236: I think this is a fair assumption, but I would reword the justification to say that deeper mineral soils are DOC sinks.

**Reply**: This hypothesis will be reformulated in the light of this suggestion. Modified in 2.3.3 section: "DOC was assumed not to be produced in the groundwater reservoir (SS), as deeper mineral horizons in soil are considered to be DOC sinks instead (Kalbitz and Kaiser, 2008) and low DOC concentrations have been observed in Kervidy-Naizin's groundwater (mean of ca. 1 mg L-1; Aubert et al. (2013))."

Line 240: why is the 'L' in [M L-3] cubed? What does the L stand for? It seems to represent units of volume, but it wouldn't make sense for that to be cubed. Please clarify and be consistent throughout the text.

**Reply:** It will be modified in the manuscript.

$[M\ L^{-3}]$ has been changed to $[M\ L^{-1}]$. L stands for a volume.

Lines 281-288: two very long sentences. Consider revising it into multiple sentences.

**Reply:** This suggestion will be taken into account in the revised manuscript.

The sentences have been split

Line 322: start new paragraph.

**Reply:** This suggestion will be taken into account in the revised manuscript.

Paragraph added

Line 324: please define which 6 metrics were used.

**Reply:** The 6 metrics used are defined in Table 3 (line 323).

Performance metrics are defined in Table 3 and more explained in the SI

Line 345: define 'Pareto front' and how to interpret.

**Reply:** 'Pareto front' will be defined and elements of interpretation will be added in the revised manuscript.

Added in 2.5 section: "When using multi-objective optimization to calibrate a model, the goal is to find a set of solutions that simultaneously optimize several, potentially conflicting, objective functions that measure individual processes. The interaction of multiple objectives leads to a set of compromised solutions known as Pareto-optimal front (hereafter, "Pareto front") (Mostafaie et al., 2018). As none of the solutions can be considered superior when there is more than one objective to optimize, Pareto-optimal solutions are also called non-dominated solutions (Yeste et al., 2023) with equally good parameter sets, which provides an uncertainty boundary of the predictive model."

Line 393: the phrase "relatively well" is vague. Be more specific.

**Reply:** We will rephrase this sentence.

As the $NSE_Q$ evaluates the models' ability to simultaneously reproduce high flows (added in SI) we have added the sentence in 3.2 section for clarification: "The high flow variations associated with storm events were usually represented relatively well ($NSE_Q > 0.75$) in calibration and evaluation periods, with good synchronicity, particularly in winter 2010 and 2014."

Lines 401-403: Consider reporting the mean and standard deviation of observed DOC and NO3 concentrations to aid RMSE interpretation. In some catchments, mean DOC and NO3 concentrations are less than or equal to your observed RMSE, so it's important to contextualize this information.

**Reply:** This suggestion will be taken into account in the revised manuscript.

The RMSE of observed DOC and NO3 have been added in the captions of Fig. 5 and Fig. 6

Lines 459-461: Great sentence!

**Reply:** Thank you

Line 478: move your key result to the beginning of this paragraph.

**Reply:** This suggestion will be taken into account in the revised manuscript.

Key result has been moved to the to the beginning of this paragraph

Line 490: move your key result to the beginning of this paragraph.

**Reply:** This suggestion will be taken into account in the revised manuscript.

Key result has been moved to the to the beginning of this paragraph

Line 491: confusing use of 'well'—is it an adverb or noun here?

**Reply:** We will clarify this sentence.

Sentence clarified in 3.4.2 section :'It also reproduced drying rates at the end of the summer and wetting rates well overall. However, the model tended to slightly underestimate soil moisture in summer 2015 and winter 2016.'

Table 2: In the DOC concentration row, Definition column, remove the word 'rate' in "DOC concentration rate…"

**Reply:** This suggestion will be taken into account in the revised manuscript.

Done

Figure 4: the way this is plotted, it's difficult to tell when the observed and simulated data are just the same or there are gaps in the data.

**Reply**: An observed vs. simulated plot with a 1:1 line will be added either inside each figure or in the SI for easier comparison between the observed and simulated data.

An observed vs. simulated streamflow plot with a 1:1 line has been added to the SI considering calibration and evaluation periods. Observations have been represented with dots rather than lines.

Figure 6: when the model predicts highest annual concentrations there is no observational data. What happened to that data? Is it missing? This needs to be explained in the methods, and some discussion is needed to explain how this might affect model accuracy.

**Reply:** These periods correspond to zero flow, where there is no observation (because there is no flowing water at the outlet). This is indeed a methodological problem in calibrating the

model for solute concentrations during these periods. The model tends to simulate relatively high nitrate concentrations during these summer periods because it does not simulate zero flow and the simulated flow is very low. These elements will be explained in the methods and discussion section.

Added in 3.2 section: The model simulated high NO3- concentrations in summer, when streamflow and NO3- concentrations had not been observed. During summer dry periods, the stream effectively dries up and no water flows at the outlet, which made it more difficult to calibrate the model to predict their solute concentrations. The model simulated near-zero water flow during dry periods, but occasionally simulated flow on certain days when zero flow had been observed, which yielded relatively high simulated NO3- concentrations. The lack of observed NO3- concentrations during dry periods also provided no constraints that could help the model represent NO3- concentrations realistically."

Figure 7: the last sentence of the caption states that "The boxplot of KGENO3 for scenarios S1 and S2 are absent because their values were negative." What do negative values mean for model performance?

**Reply:** Negative values for KGE indicate poor model performance, in this case a poor fit between observed and simulated nitrate concentration data. This precision will be added in the manuscript.

Added in caption of Fig. 7: " The boxplot of KGENO3 for scenarios S1 and S2 are not shown, as their values were negative (median = -1, for S1 and S2 scenarios). This is because the model is not calibrated to represent NO3- concentrations in these scenarios.

Figure 8: text is too small in the graphs. It might also be helpful to explain how to interpret the eCDF values somewhere in the manuscript.

**Reply:** The size of the textual parts of figure 8 will be increased. The explanation of the eCDF (the empirical cumulative distribution function of each parameter) will also be added.

The eCDF plot has been modified to a boxplot to better show the variability of the parameters values for each scenario.

Figures 9-11: The background color in the table insert makes the text illegible. Also, font size is too small.

**Reply:** The font size will be increased. The background color in the table insert will be changed.

The background table in Fig 9-10 has been deleted.

Eder A, Weigelhofer G, Pucher M, Tiefenbacher A, Strauss P, Brandl M and Blöschl G 2022 Pathways and composition of dissolved organic carbon in a small agricultural catchment during base flow conditions Ecohydrology & Hydrobiology 22 96–112

Shang P, Lu Y, Du Y, Jaffé R, Findlay R H and Wynn A 2018 Climatic and watershed controls of dissolved organic matter variation in streams across a gradient of agricultural land use Science of The Total Environment 612 1442–53

**Reply**: These references will be added in the revised manuscript

Done

References:

Anderson, T. R., Groffman, P. M., Kaushal, S. S., Walter, M. T.: Shallow groundwater denitrification in riparian zones of a headwater agricultural landscape, J. Environ. Qual., 43, 732–744, 2014. https://doi.org/10.2134/jeq2013.07.0303,.

Barton, L., C.D.A. McLay, L.A. Schipper, Smith, C.T.: Annual denitrification rates in agricultural and forest soils: A review. Aust. J. Soil Res. 37:1073–1093, 1999 doi:10.1071/SR99009

Benoit M., Veysset P.: Livestock unit calculation: a method based on energy needs to refine the study of livestock farming systems. INRAE Prod. Anim., 33, 139e-160e, 2021. https://doi.org/10.20870/productions-animales.2021.34.2.4855

Birkel, C., Soulsby, C., and Tetzlaff, D.: Integrating parsimonious models of hydrological connectivity and soil biogeochemistry to simulate stream DOC dynamics: PARSIMONIOUS COUPLED DOC MODEL, J. Geophys. Res. Biogeosci., 119, 1030–1047, 2014.

Birkel, C., Duvert, C., Correa, A., Munksgaard, N. C., Maher, D. T., and Hutley, L. B.: Tracer-Aided Modeling in the Low-Relief, Wet-Dry Tropics Suggests Water Ages and DOC Export Are Driven by Seasonal Wetlands and Deep Groundwater, Water Resour. Res., 56, 2020.

Böhlke JK et al.: Multi-scale measurements and modeling of denitrification in streams with varying flow and nitrate concentration in the upper Mississippi River basin, USA. Biogeochemistry 93:117–141, 2009.

Hefting, M.M., Bobbink, R., de Caluwe, H: Nitrous oxide emission and denitrification in chronically nitrate-loaded riparian buffer zones. J. Environ. Qual. 32:1194–1203, 2003 doi:10.2134/jeq2003.1194

Lambert, T., Pierson-Wickmann, A. C., Gruau, G., Jaffrezic, A., Petitjean, P., Thibault, J. N., and L. Jeanneau: Hydrologically driven seasonal changes in the sources and production mechanisms of dissolved organic carbon in a small lowland catchment, Water Resour. Res., 49, 5792–5803, 2013.

Lambert, T., Pierson-Wickmann, A. C., Gruau, G., Jaffrezic, A., Petitjean, P., Thibault, J. N., and L. Jeanneau: DOC sources and DOC transport pathways in a small headwater catchment as revealed by carbon isotope fluctuation during storm events, Biogeosciences, 11(11), 3043–3056, 2014.

Morel, B., Durand, P., Jaffrezic, A., Gruau, G., and Molenat, J.: Sources of dissolved organic carbon during stormflow in a headwater agricultural catchment, Hydrol. Processes, 23(20), 2888–2901, 2009.

Salmon-Monviola, J., Moreau P., Benhamou C., Durand P., Merot P., Oehler F., Gascuel-Odoux C.: Effect of climate change and increased atmospheric CO2 on hydrological and nitrogen cycling in an intensive agricultural headwater catchment in western France, Clim. Change, 120(1–2), 433–447, 2013.

---

## Author Response (AR2)

Jordy Salmon-Monviola
INRAE - UMR SAS
CS 84215
35042 Rennes Cedex

**Subject:** Submission of a revised version of a manuscript

Rennes, October 22th, 2024.

Dear editor and reviewers,

Please find enclosed a revised version of our manuscript, "Improving the hydrological consistency of a process-based solute-transport model by simultaneous calibration of streamflow and stream concentrations", written by Jordy Salmon-Monviola, Ophélie Fovet and Markus Hrachowitz.

We describe below how we have responded to the issues raised by the editor and the reviewer.

Thank you for considering the revised version of our manuscript.

Sincerely,

Jordy Salmon-Monviola on behalf of the co-authors

**Editor and Referee comments on "Improving the internal hydrological consistency of a process-based solute-transport model by simultaneous calibration of streamflow and stream concentrations" Salmon-Monviola, J., Fovet, O., and Hrachowitz, M., Hydrol. Earth Syst. Sci. Discuss. https://doi.org/10.5194/hess-2023-292, 2024.**

Editor and reviewer comments are shown in black. Authors replies are in blue.

We thank the editor and the reviewer for their very constructive comments, which help us to improve our paper. Below we outline how we have responded to the issues raised in the revised manuscript.

**Editor Comments 14 Oct 2024**

Thanks for your efforts in revising your manuscript. We have received one positive review from one of the reviews from the first round but have been waiting for another review (also from a reviewer from the first round). However, as we are now far beyond the submission time and the authors did a good job in revising the manuscript (and detailed replied to the former comments), I decided that we can move on even without the second review. There are a few minor issues as listed by the one reviewer and below, which should be addressed before publication.

We thank the editor for its positive and constructive feedback, as well as the valuable suggestions provided on this paper.

L211:"The rainfall-runoff model uses daily precipitation (P) ": please be careful whether you talk about rainfall or precipitation. I understand there is no snow in your catchment (and the model), so I suggest always using the term rainfall.

The term 'precipitation' has been changed to 'rainfall' in the manuscript and in Table A1. The term "precipitation" has been left in the text when it comes from a reference.

Appendix C presents 'common' knowledge and is, thus, not needed. Another reason for removing this part is that here, the equations are mathematically incorrect (multi-letter variable names, implying multiplication) (in the main text, the authors mainly use correct variable names, but please correct Eq 21).

Appendix C has been removed. Reference for NSE$_{FDC}$ and NSE$_{RUNOFF}$ (Sawizk et al., 2011) has been added in Table 3. Reference for PBIAS (Moriasi et al., 2007) has been added in section 3.4.1.

Eq 21 has been modified to be mathematically correct:

$$\Delta mass_{DOC_i} = Production_{DOC_i} - Loss_{DOC_i}$$

To

$$\Delta M_{DOC_i} = P_{DOC_i} - L_{DOC_i}$$

loss coefficient ($L_{DOC_i}$) (dimensionless) has been changed to $l_{DOC_i}$ in text, in Table 2 and in Fig 2.

In Table 2, to be in coherence with the text, $k_{DOC_{SU}}$ and $k_{DOC_{SUR}}$ have been modified from DOC concentration in unsaturated storage and riparian storage, to the DOC concentration at which daily DOC production occurs in unsaturated and riparian storage, respectively.

Just as a comment beyond the current study, a further alternative of an objective function that could be considered in future studies would be the modified KGE suggested by Pool et al. (2018).

Pool, S., Vis, M., & Seibert, J. (2018). Evaluating model performance: towards a non-parametric variant of the Kling-Gupta efficiency. Hydrological Sciences Journal, 63(13–14), 1941–1953. https://doi.org/10.1080/02626667.2018.1552002

Thank you for recommending the work of Pool et al. (2018). We will consider this objective function in future studies.

**RC1: 'Comment on hess-2023-292', Anonymous Referee #1, 18 Jul 2024**

In their manuscript entitled "Improving the hydrological consistency of a process-based solute-transport model by simultaneous calibration of streamflow and stream concentrations", Salmon-Monviola et al. have addressed all comments raised to my full satisfaction. I only have some minor suggestions to improve clarity and writing style, listed below. Besides these minor revisions, I am convinced it will be of high value for the readers of HESS.

We thank the reviewer for its positive and constructive comments, as well as the valuable suggestions provided on this paper.

Minor comments:

I acknowledge toning down the title and I am fine with both options. Personally, I prefer the first title starting with "Improving"

Thank you

L25: "significantly improved", not the other way around

We have corrected the sentence in the abstract.

L41-43: Sorry, but it took me three attempts to understand this sentence. Can you split it to simplify it, please?

We split this sentence

L57: Hard to follow in that order. Please rewrite to something like: "In hydrology, these insufficient model constraints can result in many equally good alternative model solutions, frequently referred to as equifinality (Beven, 2006)."

This sentence has been changed

L160: I suggest you write "opposing dynamics of" or "contrasting dynamics of", instead of "opposition of dynamics of"

This sentence has been changed

L188-190: Nested catchment is not a convincing argument for me. If you can show that soil types, slopes, and elevation are similar – that would be convincing. Otherwise, I would tone that down to "it could be indicative" instead of "it could represent"

We have revised the sentence.

"Although the Toullo station lies outside Kervidy-Naizin, we assumed that, as Kervidy-Naizin and Naizin are nested, it could represent Kervidy-Naizin's soil moisture conditions in the upland zone. "

to

"Although the Toullo station lies outside Kervidy-Naizin, we assumed that it could represent Kervidy-Naizin's soil moisture conditions in the upland zone. This assumption is supported by the fact that Kervidy-Naizin and Naizin are nested and have similar characteristics, such as soil types, slopes, and elevation (Matos-Moreira et al., 2017; Sorel et al., 2010). "

Matos-Moreira, M., Lemercier, B., Dupas, R., Michot, D., Viaud, V., Akkal-Corfini, N., Louis, B., and Gascuel-Odoux, C.: High-resolution mapping of soil phosphorus concentration in agricultural landscapes with readily available or detailed survey data, European Journal of Soil Science, 68, 281–294, https://doi.org/10.1111/ejss.12420, 2017.

Sorel, L., Viaud, V., Durand, P., and Walter, C.: Modeling spatio-temporal crop allocation patterns by a stochastic decision tree method, considering agronomic driving factors, Agricultural Systems, 103, 647–655, https://doi.org/10.1016/j.agsy.2010.08.003, 2010.

L478: I would suggest to place Figure 4 earlier (somewhat around here). Right now, it comes very late. Also, you have all the information on NSE =XX in Figure 4, right? So writing it again in parenthesis is redundant and makes reading harder. I suggest you take that out.

Figure 4 has been moved earlier. The information of metrics in parenthesis have been removed in section 3.2.

Figures 3, 5 and 6 have also been move earlier.

L846: Nice and clear conclusion

Thank you